# [RE] Are Your Models Still Fair? Fairness Attacks on Graph Neural Networks via Node Injections

## Abstract

Graph Neural Networks (GNNs) have become indispensable for learning on graph-structured data, with applications in socially sensitive domains such as recommendation systems and healthcare. However, recent research has revealed that fairness-enhancing GNNs remain vulnerable to adversarial attacks, raising concerns about their real-world robustness. This paper presents a reproducibility study of Luo et al. (2024), which demonstrates that adversarial node injection can effectively compromise fairness while preserving overall predictive accuracy. Our results confirm that such attacks are efficient (requiring minimal perturbations), realistic (exploiting feasible node injections), and deceptive (causing fairness degradation without significant accuracy loss). Along with validating the original findings, we redefine their framework as an evasion attack, showing that the attack remains effective on a clean model. Furthermore, we propose a novel defense strategy and analyze the impact of model depth on the attack. Our results highlight the need for more robust GNN architectures against fairness-targeted adversarial threats.

## 1 Introduction

Graph Neural Networks (GNNs) have significantly improved performance on tasks with graph-structured data by inherently leveraging the underlying relational information. Their ability to capture complex dependencies has led to widespread adoption in various domains centered around human interactions, such as social media prediction, recommender systems, and healthcare applications (Sankar et al., 2021; Fan et al., 2019; Yan et al., 2024; Paul et al., 2024). One of the primary applications of GNNs is node classification, where the goal is to predict labels for individual nodes based on both their attributes and graph structure (Xiao et al., 2021). Given their deployment in socially sensitive contexts, considerable research has focused on enhancing fairness in GNNs, ensuring that predictions remain unbiased across attributes like gender or race (Dai & Wang, 2021; Wang et al., 2022; Yang et al., 2024; Kose & Shen, 2024). Unfortunately, with time, we have learned that graph neural networks can be quite unfair (Chen et al., 2024), and are also vulnerable to adversarial attacks (Zügner et al., 2020; Mu et al., 2021). Existing literature has predominantly focused on degrading model utility, employing strategies that manipulate graph structure, node labels, or node features to mislead the learning process. Recently, research has begun to explore attacks aimed at undermining fairness without affecting the utility. These attacks can be hard to detect without proper fairness metrics.

Several methods have been proposed in this direction. FA-GNN (Hussain et al., 2022) introduces adversarial edge perturbations to exacerbate bias in graph-based predictions. FATE (Kang et al., 2024) employs a meta-learning-based framework to optimize adversarial manipulations that amplify fairness disparities. G-FairAttack (Zhang et al., 2024) formulates fairness attacks through a surrogate loss function while enforcing utility constraints to maintain stealthiness. Despite these advancements, existing fairness attacks primarily focus on perturbing the existing graph structure rather than injecting new nodes. This is often infeasible in real-world scenarios. However, adversaries usually do have the capability of creating new nodes, establishing connections, and setting features as they wish. This opens up a new attack surface, where fairness can be compromised by strategically injecting nodes and forming connections in a manner that undermines the model's fairness in node classification. Luo et al. (2024) propose such an attack, namely Node Injection-based Fairness Attack (NIFA), in this *gray-box* scenario by finding sensitive positions in the graph, connecting

carefully to desired nodes, and optimizing features of these nodes to undermine fairness while having only a marginal effect on the accuracy. Gray-box refers to an attack with partial knowledge of the *victim model* and the training data. The victim model refers to the target machine learning model that the attacker aims to manipulate.

In this paper, we reproduce the experiments from Luo et al. (2024) and verify their main claims. The focus of this study is specifically on group fairness with a binary sensitive attribute, such as gender or binary race categories, aligning with the original setup by Luo et al. (2024). Additionally, we address missing results and explore extensions to defense strategies for this attack. These additions include the generalization of the attack by changing it into an evasion attack, where the poisoned graph is used only during inference. Furthermore, we investigate the usage of different defense strategies, based on node injection and model architecture.

## 2 Scope of Reproducibility

The main contribution of the original paper Luo et al. (2024) is a novel gray-box poisoning attack method, namely Node Injection based Fairness Attack (NIFA). This method injects poisoned nodes based on two principles. The first of these is the *uncertainty maximization* principle, which selects the nodes in the training graph with the highest uncertainty. These nodes will serve as the target nodes on which the injected nodes will attach. The second principle is the *homophily-increase principle*. The authors define the node-level homophily-ratio $\mathcal{H}_u$ as the ratio of neighbors of $u$ that have the same sensitive attribute as the node $u$ itself. After choosing the uncertain nodes and injecting adversarial nodes, these nodes are attached to the uncertain nodes which have the same sensitive attribute as the adversarial node. The rationale behind this principle is that it will increase information propagation within the group and limit propagation across groups, which would be the case were we to add edges to uncertain nodes randomly. Finally, a surrogate network is used to optimize the features of the injected nodes to maximize the effectiveness of the attack. This attack falls under the category of *poisoning attacks*, which are attacks that occur during the training phase of the victim model and lead to poisoned models. In comparison, *evasion attacks* occur during the inference phase of the victim model and do not affect the parameters of the model. A visualization of the NIFA framework, originally from their paper Luo et al. (2024), can be seen in Figure 1.

In this study, we aim to reproduce the following claims made in Luo et al. (2024):

- **Claim 1:** NIFA can consistently attack existing GNN models with only a 1% perturbation rate, resulting in a significant increase of unfairness in the trained model.

- **Claim 2:** Utilizing NIFA results in an unnoticeable utility compromise.

- **Claim 3:** NIFA achieves state-of-the-art attack performance compared to other attacks when all methods utilize a perturbation rate of only 1%.

- **Claim 4:** The uncertainty-maximization principle, homophily-increase principle, and iterative training strategy are all needed to consistently and significantly increase unfairness of the victim model.

Perturbation rate refers to the proportion of modifications applied to the graph, such as the addition or removal of nodes or edges, relative to the graph's original structure. Utility indicates the effectiveness of the GNN in performing its intended tasks, which we measure with accuracy due to the node classification used in this study.

In addition to reproducing these claims, we also build on their research with our own research questions and study them with a series of experiments. Our added contribution answers the following questions:

- How significantly can NIFA degrade fairness when modeled as an evasion attack?

- To what extent can a node-injection strategy analogous to NIFA be used as a defense mechanism against the attack?

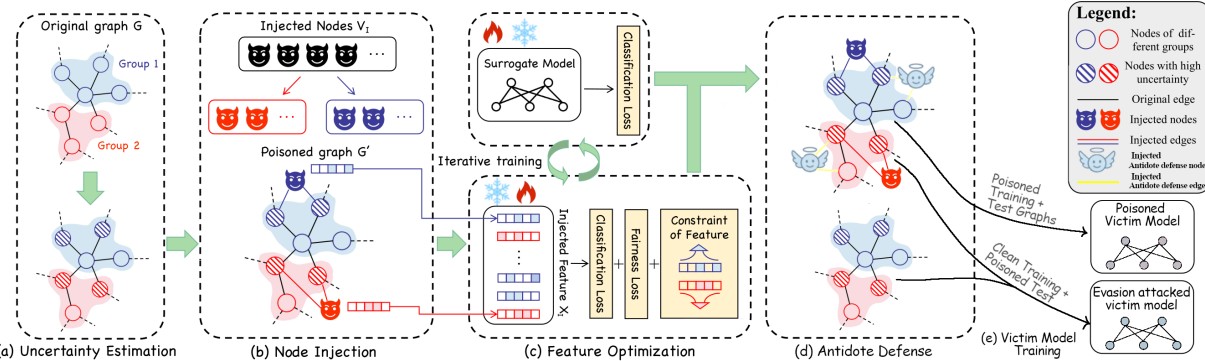

Figure 1: The complete NIFA framework from Luo et al. (2024) with our additions: Antidote defense (d) and NIFA as evasion attack (e).

- To what extent does increasing the victim model size mitigate the fairness degradation of NIFA?

While the code for the original experiments is available, independent reproduction can help in the verification of results and ensure that the experiments are not dependent on a specific setup or hyperparameters of the model, which we also explore through our extension experiments. Our study provides an independent verification of the attack's generalizability across the datasets and models used in the original paper. Additionally, it also allows the identification of potential ambiguities or mistakes in the original experimental setup. Apart from that, the ease of reproducibility is also a concern, as the availability of code does not guarantee a straightforward and successful replication of results. Specifically, missing hyperparameters can prompt a grid search, revealing unexpected results. Furthermore, missing code for fairness-aware models in the repository can require significant pre-processing. Difficulty in reproducing the results can also indicate inconsistencies in datasets, methodology, or the attack itself, something our extension experiments provide insights into.

## 3 Methodology

### 3.1 Model Descriptions

The paper uses multiple victim models to study the effectiveness of the attacks. These are as follows:

- **GCN** (Kipf & Welling, 2017): GCN is a standard graph convolutional network, borrowing the concept of convolution from computer vision.

- **GraphSAGE** Hamilton et al. (2017): GraphSAGE samples a fixed number of neighbors at each layer to avoid neighborhood explosion. This improves the training efficiency as well.

- **APPNP** (Gasteiger et al., 2022): A GNN that combines graph convolutions with personalized PageRank (Page et al., 1999) and decouples the prediction and propagation in the training process.

- **SGC** (Wu et al., 2019): SGC is a variant of GCN which removes the non-linearities from GCN and essentially collapses the GNN into a single matrix multiplication.

Furthermore, for increased generalizability we evaluate the NIFA framework on:

- **GAT** (Velickovic et al., 2017): GAT is the vanilla graph attention network, leveraging self-attention to enable specifying different weights to different neighbors' features.

In addition to these above, we also use 3 fairness-aware (Chen et al., 2024) victim models. These are models which incorporate techniques for reducing disparities and mitigating biases across different sensitive attributes.

- **FairGNN** (Dai & Wang, 2021): It uses a sensitive attribute estimator to predict sensitive information and an adversarial learning module to minimize the correlation between node representations and sensitive attributes.

- **FairVGNN** (Wang et al., 2022): It is a network that identifies and masks channels highly correlated with sensitive attributes to tackle the problem of sensitive attribute leakage.

- **FairSIN** (Yang et al., 2024): It is a novel approach which works by introducing additional features into the node representations instead of filtering out information.

All these models are used in the replication of the work of the original paper. However, in our experiments to answer the additional question we propose, some of them are omitted. The models not used in any experiment will be specified in the results for the corresponding experiment.
In addition, the following models are used for executing NIFA:

- **Bayesian GNN**: For uncertainty estimation, a Bayesian GNN is used with Monte Carlo dropout. Given a 2-layer GCN the model parameters are estimated using $T$ times independent Bernoulli dropout sampling. Using these models, $T$ different predictions are made, resulting in a per node variance. This variance positively correlates with model uncertainty.

- **Surrogate GNN**: NIFA uses a surrogate GNN for optimizing the features of injected nodes. This uses an objective function devised by the authors.

### 3.2 Baselines

To compare the effectiveness of NIFA, three different state-of-the-art fairness gray-box graph attacks were conducted. FA-GNN (Hussain et al., 2022), FATE (Kang et al., 2024), and G-FairAttack (Zhang et al., 2024). For FA-GNN, the DD strategy is employed, as it was demonstrated to be more effective in the original study. For all baselines, the number of injected nodes or modified edges was set to be equal to those in NIFA. Similar to the original research, these attacks are conducted on the GCN victim model.

### 3.3 Datasets

We use three real-world datasets, namely Pokec-z, Pokec-n, and DBLP. For all datasets, we use 50% of the labelled nodes for training, 25% for validation, and 25% for test, identical to the original paper. **Pokec-z** and **Pokec-n** are subgraph samples from Pokec, a popular Slovak social-network (Dai & Wang, 2021). The subgraphs correspond to

| Dataset | Pokec-z | Pokec-n | DBLP |
|---|---|---|---|
| # of nodes | 67,796 | 66,569 | 20,111 |
| # of edges | 617,958 | 517,047 | 57,508 |
| Feature dim. | 276 | 265 | 2,530 |
| # of labeled nodes | 10,262 | 8,797 | 3,196 |

Table 1: Dataset Statistics as used in our experiments.

different regions. Each node represents a user, while edges represent a unidirectional following relationship. The node features include profile information like age, gender, hobbies etc. and the classification task is to predict the working field of users.
**DBLP** is a bibliographical dataset which represents academic collaboration networks (Hussain et al., 2022). The nodes in this dataset are authors, and edges represent co-authorship. Node features are based on keywords present in authors' publications. If two authors have published a paper together, they will be connected. The classification task here is to predict the research area of the authors. Detailed statistics are summarized in Table 1. We would like to mention that in case of DBLP, while the number of labeled nodes mentioned is correct, less than 1300 are used in the training, validation, and testing. Some of the labeled nodes have a missing sensitive attribute and hence are not used. This discrepancy is further elaborated on in Section 5.5. Due to this discrepancy, we also use the dataset from Hussain et al. (2022), the paper that adapted this dataset for training of fairness models. We ran experiments on this dataset, of which the results are shown in Appendix C. The main results we display are from the dataset as supplied in the authors' repository to keep the experimental conditions as close as possible to the original paper.

### 3.4 Hyperparameters

In reproducing the original experiments, we use the same hyperparameter settings as the original paper, which are summarized below.

**Victim Models**: For all victim models, the hidden layer size is set to 128 and the learning rate is set to 0.001. For most victim models, dropout is set to 0, except in the case of FairGNN and FairVGNN, where 0.5 is used instead. The layer number is 2 for GCN and GraphSAGE. For APPNP, the teleport probability $\alpha$ is set to 0.2 and iteration number $k$ is set to 0. FairGNN uses GAT as backbone, while the FairVGNN uses GCN. For FairVGNN, the prefix cutting threshold $\epsilon$ is searched from $\{0.01, 0.1, 1\}$, and mask density $\alpha$ is set as 0.5. The epochs for the generator, discriminator, and classifier are searched from $\{5, 10\}$. This is also in accordance with the original paper, and it does mention exactly which values are used for the reported results. However, the hyperparameters with which we obtained the results closest to the paper are mentioned in Table 2. FairSIN also uses the GCN as backbone, and the weight of neutralized feature $\delta$ is set as 4.

| Notations | Pokec-z | Pokec-n | DBLP |
|---|---|---|---|
| $\epsilon$ | 1.0 | 1.0 | 1.0 |
| $d$ | 10 | 5 | 10 |
| $g$ | 10 | 5 | 5 |
| $c$ | 5 | 10 | 5 |
| epochs | 1000 | 1000 | 500 |

Table 2: Hyperparameters for final FairVGNN results. $d$, $g$, and $c$ refer to the epochs for discriminator, generator, and classifier, respectively.

**NIFA**: For NIFA, the hyperparameters that are dataset dependent and are shown in Table 3. $\alpha$ and $\beta$ are the weights of the components of the objective function. $\bar{d}$ is the degree of injected nodes, and $b$ is the constraint on the number of injected nodes. $b$ is set to 1%, but since dataset sizes are different, the absolute value is also different. $\bar{d}$ represents the average node degrees in the datasets. $k$ is the uncertainty threshold, which is the fraction of uncertain nodes that serve as candidates for injection points. $max\_iter$ is the number of rounds the optimizer runs for, and $max\_step$ is the number of epochs per round. Apart from these, the learning rate for optimizing the surrogate model and node features is 0.001, and the dropout ratio is set to 0. $T$ is the sampling times of the Bayesian Network and is set to 20. The model itself is a two-layer GCN with hidden size of 128.

| Notations | Pokec-z | Pokec-n | DBLP |
|---|---|---|---|
| $\alpha$ | 0.01 | 0.01 | 0.1 |
| $\beta$ | 4 | 4 | 8 |
| $b$ | 102 | 87 | 32 |
| $\bar{d}$ | 50 | 50 | 24 |
| $k$ | 0.5 | 0.5 | 0.5 |
| max_step | 50 | 50 | 50 |
| max_iter | 20 | 20 | 10 |
| epochs | 1000 | 1000 | 500 |

Table 3: Hyperparameters used for NIFA.

### 3.5 Utility and Fairness Metrics

We use the following metrics for evaluating our models. For the fairness metrics, we focus on group fairness, with a binary sensitive attribute $s \in \{0, 1\}$, in correspondence with previous works (Dai & Wang, 2021; Dong et al., 2022; Ling et al., 2023; Luo et al., 2024).

1. Accuracy: This is the percentage of correctly classified nodes, which serves as the utility metric.

2. $\Delta_{SP}$ : This metric is derived from statistical parity, which requires the predictions to be independent of the sensitive attribute. Formally, statistical parity is defined as:

$$P(\hat{y}_v = y | s = 0) = P(\hat{y}_v = y | s = 1) \tag{1}$$

From this, our first fairness metric can be defined as:

$$\Delta_{SP} = \mathbb{E}|P(\hat{y} = y | s = 0) - P(\hat{y} = y | s = 1)| \tag{2}$$

3. $\Delta_{EO}$: This metric comes from the idea of equal opportunity, which required the predicting correctly to be independent of the sensitive attribute. Formally,

$$P(\hat{y}_v = y | y_v = y, s = 0) = P(\hat{y}_v = y | y_v = y, s = 1) \tag{3}$$

From this, our second fairness metric can be defined as:

$$\Delta_{EO} = \mathbb{E}|P(\hat{y} = y | y = y, s = 0) - P(\hat{y} = y |, y = y, s = 1)| \tag{4}$$

### 3.6 Experimental setup and code

To evaluate the effectiveness of NIFA, baselines were obtained by running the models on clean graphs. To account for the stochastic nature of the process, every model was run 5 times on each dataset. We report the mean along with the standard deviation of our results obtained. After setting these baselines, the graph was poisoned using NIFA and the models were re-trained on these graphs to get the final attack results, which comprise the three metrics listed in Section 3.5. Apart from this, we made some changes to the code to run our experiments. The code, and the instructions to run these experiments, can be found on https://anonymous.4open.science/r/fact-ai-8CEB/README.md. All our experiments can be replicated from the command line without any changes to the code. The used packages are slightly different from the original research, and corresponding versions can be found in the environment files in the repository.

### 3.7 Computational Requirements

The experiments were done using an Intel Xeon Platinum 8360Y CPU, an Nvidia A100 GPU and 128 GB of RAM. Each model run of the main reproducibility results took around 15 minutes to run. The experiment results we report are from 5 iterations run on 3 datasets. Approximate runtimes for all experiments can be deduced from this. The Python and module versions can be found in our GitHub repository.

## 4 Results

### 4.1 Results reproducing original claims

The results of replicating the original experiments are shown in Table 4. These results correspond to the original claims we listed earlier about the paper.

**Claim 1**: From the results found in Table 4, we can see that NIFA does indeed consistently increase unfairness in GNN models while using only a 1% perturbation rate. Therefore, we can say our results support this claim. The attack effectiveness depends on the model and the dataset, but the attacks consistently give us a less fair model. However, the fairness-aware models do not exhibit a consistently significant increase in their parity or equality values before and after the attack. Only on the Pokec-n dataset does FairGNN show both a parity and equality metric increase, whereas on the other datasets only one of the metrics increases. However, we note that our results for the fairness-aware models differ somewhat from the original results obtained in the paper. This might be caused by the difference in implementation, as the authors did not release code for these models in their repository. Notably, on the FairSIN model, the NIFA attack significantly decreases fairness metrics on the DBLP dataset, while it does not cause significant changes on the Pokec datasets. Furthermore, we note different effectiveness of NIFA on the DBLP dataset, compared to the Pokec dataset. We hypothesize that due to significantly lower degree of the former dataset compared to the latter, NIFA might be more effective. Results substantiating this claim can be found in Appendix D.

**Claim 2**: From the accuracies in the table, we can see that accuracies before and after the attack are quite similar, especially for the Pokec datasets, where it is mostly within 1%. However, the drop is slightly higher for DBLP, showing around a 5% decrease in accuracy. This results in an attack which is unnoticable up to 5% percent utility decrease.

**Claim 3**: Comparing to other attack methods, in Table 5, NIFA increases unfairness significantly across all datasets. Confirming the claim made by the authors of Luo et al. (2024).

Table 4: Our Accuracy and Fairness performance of NIFA on different victim models compared to the reference found in Luo et al. (2024). The results are reported in percentage (%). The colored values (+/-) indicate the difference between our and the original author's results. The **Bold** values indicate a significant increase in $\Delta_{SP}$ or $\Delta_{EO}$ after poisoning.

| | | Pokec-z | | | Pokec-n | | | DBLP | | |
|---|---|---|---|---|---|---|---|---|---|---|
| | | Accuracy | $\Delta_{SP}$ | $\Delta_{EO}$ | Accuracy | $\Delta_{SP}$ | $\Delta_{EO}$ | Accuracy | $\Delta_{SP}$ | $\Delta_{EO}$ |
| GCN | before | 71.50 +0.28 | 7.84 +0.71 | 5.87 +0.77 | 70.34 -0.58 | 2.23 +1.35 | 2.33 -0.11 | 96.58 +0.70 | 4.36 +0.52 | 2.42 +0.51 |
| | after | 70.76 +0.26 | **17.22** -0.14 | **15.43** -0.16 | 70.21 +0.09 | **8.93** -1.17 | **8.50** -1.35 | 93.47 +0.10 | **11.19** -2.30 | **17.75** -2.58 |
| GraphSAGE | before | 70.70 -0.09 | 4.49 +0.20 | 3.43 -0.03 | 69.14 +0.37 | 0.96 -0.69 | 1.39 -0.26 | 96.23 -0.35 | 2.64 -1.63 | 2.39 -0.39 |
| | after | 70.16 +0.11 | **5.60** -0.60 | **3.96** -0.24 | 68.43 0.00 | **0.98** -2.34 | **2.44** -1.12 | 93.62 -2.84 | **9.62** -0.54 | **16.60** -0.05 |
| APPNP | before | 69.88 +0.09 | 7.26 +0.43 | 5.53 +0.46 | 68.33 -0.40 | 4.40 +1.01 | 4.73 +1.02 | 96.58 0.00 | 3.97 -0.01 | 2.58 +0.38 |
| | after | 68.63 -0.49 | **17.85** -0.59 | **16.39** -0.46 | 67.90 0.00 | **9.77** -3.70 | **9.53** -3.99 | 91.51 -0.95 | **13.20** -0.68 | **17.30** -2.90 |
| SGC | before | 68.37 -0.72 | 5.54 -1.74 | 3.85 -1.60 | 66.48 -0.47 | 4.64 +1.90 | 4.96 +1.75 | 96.63 0.00 | 4.70 0.00 | 3.25 +0.14 |
| | after | 67.19 -0.64 | **16.23** -1.42 | **14.71** -1.38 | 64.32 -2.40 | **7.51** -3.08 | **7.36** -3.31 | 92.31 -1.57 | **10.25** -3.63 | **15.44** -4.81 |
| FairGNN | before | 70.27 +1.52 | 0.94 -0.95 | 2.98 +1.47 | 69.66 +0.25 | 3.79 +2.37 | 5.97 +3.65 | 90.70 -2.42 | 2.56 +0.61 | 0.32 -2.77 |
| | after | 69.81 +0.43 | **3.47** -2.24 | 1.41 -2.81 | 68.98 -0.69 | **10.48** +4.35 | **9.08** +2.75 | 90.60 -1.96 | 2.35 -3.54 | **2.50** -7.98 |
| FairVGNN | before | 64.67 -3.9 | 2.76 -1.03 | 2.65 +0.06 | 65.42 -2.35 | 2.96 +1.06 | 2.35 -0.75 | 90.30 -4.88 | 1.42 -0.48 | 3.14 +0.23 |
| | after | 63.44 -4.21 | **7.01** -4.0 | **5.79** -3.49 | 64.41 -1.33 | 4.01 +0.50 | 3.88 +0.23 | 86.78 -4.78 | **9.83** +1.87 | **18.27** +4.70 |
| FairSIN | before | 65.80 -1.53 | 2.26 +0.53 | 2.62 +0.01 | 65.25 -1.93 | 6.27 +5.88 | 7.98 +5.58 | 92.81 -1.91 | 3.35 +3.12 | 11.47 +11.02 |
| | after | 61.70 -4.85 | 1.88 -7.60 | 2.33 -8.06 | 63.64 -2.56 | 5.59 -6.23 | 5.93 -8.65 | 87.84 -4.62 | **12.11** +1.21 | **25.96** +2.31 |
| GAT | before | 71.26 – | 7.94 – | 5.89 – | 69.88 – | 1.00 – | 1.75 – | 94.97 – | 3.20 – | 3.32 – |
| | after | 71.08 – | 8.97 – | 6.93 – | 69.78 – | 1.92 – | 2.79 – | 95.63 – | 3.00 – | 5.92 – |

Table 5: Our accuracy and Fairness performance runs of the baselines compared to our NIFA run, both conducted on GCN. Results are reported in percentage (%). The colored values (+/-) indicate the difference between our and the original author's results. The missing values of FATE and G-FairAttack were not obtained due to High memory usage.

| | Pokec-z | | | Pokec-n | | | DBLP | | |
|---|---|---|---|---|---|---|---|---|---|
| | Accuracy | $\Delta_{SP}$ | $\Delta_{EO}$ | Accuracy | $\Delta_{SP}$ | $\Delta_{EO}$ | Accuracy | $\Delta_{SP}$ | $\Delta_{EO}$ |
| Clean | 71.50 +0.28 | 7.84 +0.71 | 5.87 +0.77 | 70.34 -0.58 | 2.23 +1.35 | 2.33 -0.11 | 96.58 +0.70 | 4.36 +0.52 | 2.42 +0.51 |
| FA-GNN | 71.72 +1.92 | 8.33 +1.71 | 13.23 +4.56 | 70.19 -0.61 | 4.00 +1.36 | 5.79 +2.34 | 95.43 -0.05 | 1.56 -1.76 | 4.78 -3.96 |
| FATE | - | - | - | - | - | - | 90.05 -4.82 | 2.43 -1.19 | 8.18 +6.06 |
| G-Fair | - | - | - | - | - | - | 94.02 -1.10 | 4.6 -2.20 | 13.86 +10.92 |
| NIFA | 70.76 +0.26 | **17.22** -0.14 | **15.43** -0.16 | 70.21 +0.09 | **8.93** -1.17 | **8.50** -1.35 | 93.47 +0.10 | **11.19** -2.30 | **17.75** -2.58 |

## 4.2 Results reproducing the ablation study

Reproducing the ablation study showed significant consistency. The different principles are left from the framework and the attack is tested on the GCN model. Similarly to the original authors, 3 new variants are considered. 1) *NIFA-U*: the uncertainty maximization principle is removed, and target nodes are sampled randomly. 2) *NIFA-H*: The homophily-increase is removed, meaning that each injected node may connect to targeted nodes with a different sensitive group. 3)

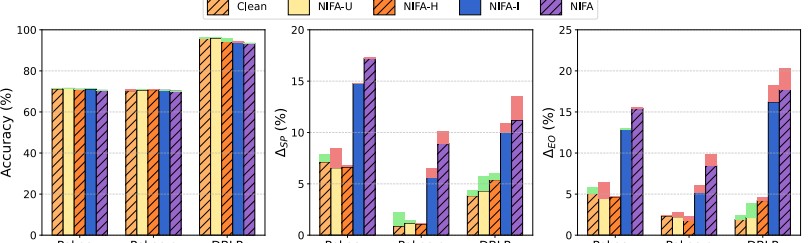

Figure 2: Ablation study of each principle in NIFA. The green values indicate an increase of a specific value in our results, compared to the original authors' results. The red values indicate the decrease of our metric compared to the original authors' results.

*NIFA-I*: The iterative training strategy is removed, meaning that the victim model is trained first, followed by the training of the injected feature matrix.

**Claim 4:** From the results in Figure 2, we can see that the trend of the results matches the original paper. This confirms their claim, that to significantly and consistently improve the effectiveness of the attack, all steps are necessary.

Our accuracies are similar to the original paper, but some small differences exist in the fairness metrics. Our *NIFA-I* and *NIFA* values are lower than the original study for Pokec-n and DBLP, while they are the

same for Pokec-z. *NIFA-H* for the most part is similar to the original results, and the difference in *NIFA-U* is dataset-dependent. However, as stated before, the trend across different ablations is consistent with the trend in the original paper.

### 4.3 Extension Experiments

**NIFA as an Evasion Attack**: We modeled NIFA as an evasion attack by training the victim models on a clean graph. We used all the models except FairGNN, FairVGNN and FairSIN for this experiment. This attack uses the same amount of injected nodes and edges as NIFA, with the only change being the training graph; the training graph is not poisoned in the evasion attack. Our results are summarized in Table 6. We see from the results that a node-injection attack using the principles of NIFA can effectively achieve the same effect on the model even if the model itself is not trained on the poisoned graph. We believe this is because of how feature-optimization works in NIFA. It is designed to mislead the model in the classification of good nodes. During the classification (or inference) phase, the target node receives this misleading information during the aggregation of features from the neighborhood. As a result, our node is exposed to adversarial features during message passing, regardless of whether the model is being trained on the poisoned graph. However, we also notice that in the case of DBLP, the drop in accuracy observed is higher for the evasion attack compared to the poisoning attack, making the evasion attack more vulnerable to detection by simpler utility-based metrics. However, we believe that in a real-world scenario, the evasion attack is a more practical scenario because it allows the attacker to perform the fairness attack at any point in time. Although GAT was not evaluated in the original paper, our results show that the NIFA poisoning attack had little effect on its fairness—for example, on DBLP, $\Delta_{SP}$ and $\Delta_{EO}$ changed only slightly from 3.20/3.32 in the clean setting to 3.00/5.92 under poisoning—whereas the evasion attack dramatically increased these gaps (with $\Delta_{SP}/\Delta_{EO}$ surging to 11.02/16.64 on DBLP, 26.24/24.82 on Pokec-z, and 24.82/24.56 on Pokec-n), demonstrating its potent efficacy as an attack strategy.

Table 6: The NIFA evasion attack compared to the original NIFA poisoning attack, and no attack at all. The **Bold** values indicate a significant increase in $\Delta_{SP}$ or $\Delta_{EO}$ over the poisoning version of NIFA.

| | | Pokec-z | | | Pokec-n | | | DBLP | | |
|---|---|---|---|---|---|---|---|---|---|---|
| | | Accuracy | $\Delta_{SP}$ | $\Delta_{EO}$ | Accuracy | $\Delta_{SP}$ | $\Delta_{EO}$ | Accuracy | $\Delta_{SP}$ | $\Delta_{EO}$ |
| **GCN** | Clean | 71.50±0.14 | 7.84±1.07 | 5.87±1.04 | 70.34±0.71 | 2.23±1.25 | 2.33±1.33 | 96.58±0.49 | 4.36±1.22 | 2.42±1.03 |
| | Poisoning | 70.76±0.39 | 17.22±0.95 | 15.43±1.03 | 70.21±0.68 | 8.93±2.25 | 8.50±2.23 | 93.47±0.48 | 11.19±2.33 | 17.75±2.44 |
| | Evasion | 70.80±0.25 | **22.50±1.60** | **20.79±1.68** | 70.21±0.40 | **11.79±3.79** | **11.79±3.79** | 91.76±0.58 | **15.08±2.95** | **22.02±3.30** |
| **GraphSAGE** | Clean | 70.70±0.66 | 4.49±1.31 | 3.43±0.95 | 69.14±0.87 | 0.96±0.67 | 1.39±0.71 | 96.23±0.28 | 2.64±1.73 | 2.39±1.03 |
| | Poisoning | 70.16±0.47 | 5.60±2.11 | 3.96±1.87 | 68.43±0.30 | 0.98±0.78 | 2.44±1.74 | 93.62±0.44 | 9.62±3.29 | 16.60±3.31 |
| | Evasion | 69.07±0.53 | **18.97±5.31** | **17.38±5.37** | 67.47±0.50 | **17.81±6.15** | **17.39±6.15** | 90.50±1.42 | **17.00±2.63** | **24.15±2.65** |
| **APPNP** | Clean | 69.88±0.70 | 7.26±0.48 | 5.53±0.52 | 68.33±0.49 | 4.40±0.65 | 4.73±0.65 | 96.58±0.34 | 3.97±1.56 | 2.58±1.17 |
| | Poisoning | 68.63±1.18 | 17.85±1.63 | 16.39±1.68 | 67.90±0.64 | 9.77±1.26 | 9.53±1.23 | 91.51±0.64 | 13.20±4.21 | 17.30±4.95 |
| | Evasion | 68.72±0.69 | **21.55±3.59** | **20.11±3.73** | 68.15±0.38 | **12.32±4.43** | **11.96±4.44** | 87.79±2.15 | **19.95±2.06** | **26.30±1.49** |
| **SGC** | Clean | 68.37±1.30 | 5.54±1.40 | 3.85±1.19 | 66.48±2.35 | 4.64±2.03 | 4.96±1.98 | 96.63±0.38 | 4.70±0.93 | 3.25±0.72 |
| | Poisoning | 67.19±0.89 | 16.23±1.32 | 14.71±1.32 | 64.32±2.67 | 7.51±1.17 | 7.36±1.30 | 92.31±1.54 | 10.25±2.31 | 15.44±2.13 |
| | Evasion | 67.08±0.89 | **23.07±2.68** | **21.66±2.68** | 66.91±0.33 | **14.66±3.82** | **14.28±3.82** | 86.93±1.78 | **20.64±1.79** | **26.80±1.19** |
| **GAT** | Clean | 71.26±0.40 | 7.94±0.75 | 5.89±0.74 | 69.88±0.27 | 1.00±0.44 | 1.75±0.87 | 94.97±0.76 | 3.20±2.84 | 3.32±1.20 |
| | Poisoning | 71.08±0.52 | 8.97±1.98 | 6.93±2.04 | 69.78±0.49 | 1.92±1.16 | 2.79±1.05 | 95.63±1.29 | 3.00±1.49 | 5.92±3.37 |
| | Evasion | 69.47±0.57 | **26.24±10.17** | **24.82±10.46** | 67.85±0.85 | **24.82±4.52** | **24.56±4.51** | 92.01±3.42 | **11.02±6.69** | **16.64±7.68** |

**Antidote Defense:** We propose and evaluate a defense strategy against NIFA. NIFA's feature optimization accentuates the unfairness of the models by minimizing an objective function. We attempt to counter this problem by injecting good nodes and optimizing the features such that they induce fairness into our graph. That is, we optimize a different fairness objective function during feature optimization compared to Luo et al. (2024):

$$L_{AD} = L_{CE} + \alpha L_{CF} - \beta(L_{SP} + L_{EO})$$

where $L_{CE}$ is cross-entropy loss, $L_{CF}$ is constraint of feature and $L_{SP}$ and $L_{EO}$ are fairness loss terms for statistical parity and equal opportunity. For exact definitions of these terms refer to Luo et al. (2024).

This approach is analogous to how NIFA itself works. Instead of optimizing for unfairness, we inject more nodes and optimize them for fairness. This defense strategy increases the number of nodes and edges and may significantly slow down some models depending on the model architecture and depth. Additionally, feature

optimization is also an overhead but remains small when compared to the training of the victim model itself. However, for the 2-layer models that we used, we did not notice a significant change in runtimes in training and testing. Specifically, we inject 300 nodes, with each node having a budget of 100 edges. The top 75% of most uncertain nodes are considered as nodes to attach to the injected good nodes. The fairness- and classifier loss are weighted by factors $\alpha = 1$ and $\beta = 1$ respectively. This is compared to reproduced results for the masking defense proposed in the original paper. For the masking defense, we mask 60% of the uncertain nodes as it is the value that gives the greatest reduction in fairness metrics according to the original paper. Our results, summarized in Table 7 show that it consistently manages to reduce $\Delta_{SP}$ and $\Delta_{EO}$, in some cases doing even better than the cleanly trained model itself. Additionally, for most of the cases, the change in accuracy is also insignificant, which means that this defense mechanism does not impact the utility of the model. SGC is an outlier in this aspect, showing a decrease in accuracy of around 5.28% when comparing with and without Antidote defense on DBLP. Using the Antidote defense on the evasion attack variant of NIFA, we observe some mixed results and can be found in Appendix F.

Table 7: Accuracy and Fairness metrics comparing the Antidote defense and node masking defense against a clean graph and a poisoned graph without the defense. The **Bold** values indicate a significant decrease in $\Delta_{SP}$ or $\Delta_{EO}$ after using the defense.

| | | Pokec-z | | | Pokec-n | | | DBLP | | |
| --- | --- | --- | --- | --- | --- | --- | --- | --- | --- | --- |
| | | Accuracy | $\Delta_{SP}$ | $\Delta_{EO}$ | Accuracy | $\Delta_{SP}$ | $\Delta_{EO}$ | Accuracy | $\Delta_{SP}$ | $\Delta_{EO}$ |
| GCN | Clean | 71.50±0.14 | 7.84±1.07 | 5.87±1.04 | 70.34±0.71 | 2.23±1.25 | 2.33±1.33 | 96.58±0.49 | 4.36±1.22 | 2.42±1.03 |
| | No defense | 70.76±0.39 | 17.22±0.95 | 15.43±1.03 | 70.21±0.68 | 8.93±2.25 | 8.50±2.23 | 93.47±0.48 | 11.19±2.33 | 17.75±2.44 |
| | Masking | 69.91±0.37 | 13.65±2.43 | 11.83±2.40 | 69.68±0.46 | 7.16±5.58 | 7.36±5.34 | 90.95±0.75 | 11.28±3.24 | 16.65±3.99 |
| | Antidote | 70.11±0.53 | **4.33±2.66** | **3.58±1.15** | 69.99±0.67 | **0.86±0.88** | **3.06±1.02** | 92.51±0.92 | 11.78±6.96 | **10.70±5.47** |
| GraphSAGE | Clean | 70.70±0.66 | 4.49±1.31 | 3.43±0.95 | 69.14±0.87 | 0.96±0.67 | 1.39±0.71 | 96.23±0.28 | 2.64±1.73 | 2.39±1.03 |
| | No defense | 70.16±0.47 | 5.60±2.11 | 3.96±1.87 | 68.43±0.30 | 0.98±0.78 | 2.44±1.74 | 93.62±0.44 | 9.62±3.29 | 16.60±3.31 |
| | Masking | 67.90±0.58 | 10.54±2.81 | 8.85±2.81 | 66.07±0.89 | 9.93±2.81 | 9.73±3.04 | 88.39±2.03 | 9.80±6.11 | 14.40±8.17 |
| | Antidote | 70.16±0.28 | **4.52±1.67** | 3.16±0.91 | 68.24±0.67 | 1.90±2.20 | 4.18±1.29 | 91.61±0.89 | **2.33±1.18** | **4.09±2.63** |
| APPNP | Clean | 69.88±0.70 | 7.26±0.48 | 5.53±0.52 | 68.33±0.49 | 4.40±0.65 | 4.73±0.65 | 96.58±0.34 | 3.97±1.56 | 2.58±1.17 |
| | No defense | 68.63±1.18 | 17.85±1.63 | 16.39±1.68 | 67.90±0.64 | 9.77±1.26 | 9.53±1.23 | 91.51±0.64 | 13.20±4.21 | 17.30±4.95 |
| | Masking | 67.25±1.10 | 14.66±3.44 | 13.19±3.58 | 67.26±0.57 | 6.41±2.60 | 6.23±2.74 | 89.45±0.94 | 8.26±3.40 | 13.63±4.54 |
| | Antidote | 66.87±1.10 | **1.55±2.33** | **3.15±0.98** | 67.55±0.62 | **4.24±2.16** | **4.55±2.17** | 88.84±1.76 | **4.64±2.23** | **9.07±2.71** |
| SGC | Clean | 68.37±1.30 | 5.54±1.40 | 3.85±1.19 | 66.48±2.35 | 4.64±2.03 | 4.96±1.98 | 96.63±0.38 | 4.70±0.93 | 3.25±0.72 |
| | No defense | 67.19±0.89 | 16.23±1.32 | 14.71±1.32 | 64.32±2.67 | 7.51±1.17 | 7.36±1.30 | 92.31±1.54 | 10.25±2.31 | 15.44±2.13 |
| | Masking | 65.82±1.46 | 11.95±3.72 | 10.51±3.64 | 65.22±1.06 | 9.93±1.54 | 9.76±1.62 | 89.25±1.21 | 9.58±3.26 | 15.17±3.76 |
| | Antidote | 66.87±0.81 | **5.81±2.23** | **4.20±2.12** | 63.82±3.66 | **3.33±3.60** | **5.24±2.47** | 87.44±2.00 | **4.26±1.08** | **8.35±2.01** |

Table 8 compares the performance of the Antidote defense against Masking and no defense on NIFA and all baselines. The Antidote defense achieves comparable performance on G-Fair and FA-GNN, with the exception of the DBLP dataset in FA-GNN. Furthermore, FATE achieves significantly higher $\Delta_{SP}$ and $\Delta_{EO}$ when either the Masking or Antidote defense is used.

Table 8: Accuracy and Fairness metrics comparing the Antidote defense and node masking defense against a clean graph and a poisoned graph without the defense, on NIFA and all baselines. The **Bold** values indicate the lowest $\Delta_{SP}$ and $\Delta_{EO}$ for every dataset.

| | Pokec-z | | | Pokec-n | | | DBLP | | |
| --- | --- | --- | --- | --- | --- | --- | --- | --- | --- |
| | Accuracy | $\Delta_{SP}$ | $\Delta_{EO}$ | Accuracy | $\Delta_{SP}$ | $\Delta_{EO}$ | Accuracy | $\Delta_{SP}$ | $\Delta_{EO}$ |
| Clean | 71.50±0.14 | 7.84±1.07 | 5.87±1.04 | 70.34±0.71 | 2.23±1.25 | 2.33±1.33 | 96.58±0.49 | 4.36±1.22 | 2.42±1.03 |
| NIFA | 70.76±0.39 | 17.22±0.95 | 15.43±1.03 | 70.21±0.68 | 8.93±2.25 | 8.50±2.23 | 93.47±0.48 | 11.19±2.33 | 17.75±2.44 |
| + Masking | 69.91±0.37 | 13.65±2.43 | 11.83±2.40 | 69.68±0.46 | 7.16±5.58 | 7.36±5.34 | 90.95±0.75 | 11.28±3.24 | 16.65±3.99 |
| + Antidote | 70.11±0.53 | **4.33±2.66** | **3.58±1.15** | 69.99±0.67 | **0.86±0.88** | **3.06±1.02** | 92.51±0.92 | 11.78±6.96 | **10.70±5.47** |
| FA-GNN | 71.72 ± 0.77 | 8.33 ± 2.60 | 13.23 ± 5.18 | 70.19 ± 0.42 | 4.00 ± 2.17 | 5.79 ± 1.52 | 95.43 ± 00.89 | 1.56 ± 0.83 | 4.78 ± 2.74 |
| + Masking | 70.15±0.37 | 9.17±1.25 | 7.18±1.40 | 69.70±0.65 | **1.74±0.82** | **3.11±0.59** | 94.97±0.16 | **1.74±0.49** | **1.60±0.77** |
| + Antidote | 69.83±0.61 | **2.19±1.65** | **2.64±0.72** | 69.35±0.53 | 2.53±0.53 | 3.56±0.98 | 93.62±1.33 | 16.56±8.28 | 14.33±5.40 |
| FATE | - | - | - | - | - | - | 90.05±0.26 | **2.43±0.45** | **8.18±0.73** |
| + Masking | - | - | - | - | - | - | 44.67±10.20 | 9.78±14.25 | 11.43±13.53 |
| + Antidote | - | - | - | - | - | - | 84.67±1.32 | 18.26±3.85 | 26.29±5.34 |
| G-Fair | - | - | - | - | - | - | 94.02±0.14 | 4.6±1.02 | 13.86±1.05 |
| + Masking | - | - | - | - | - | - | 93.17±0.77 | **3.25±2.31** | **7.50±3.83** |
| + Antidote | - | - | - | - | - | - | 94.92±0.66 | 5.71±3.74 | 10.73±5.12 |

**Deeper Victim Models:** As current research is focused on fairness attacks of quite shallow GNNs with only 1 to 2 layers (Song & Palanisamy, 2024; Luo et al., 2024; Dai & Wang, 2021; Hussain et al., 2022),

Table 9: NIFA performance when increasing the number of layers of the victim model. For SGC, only the hop-number is increased. The 4 layer FairGNN model uses a gelu activation function, kaiming initialization, layer normalization, and residual connections. The **bold** values indicate a significant increase of fairness.

| | | Pokec-z | | | Pokec-n | | | DBLP | | |
|---|---|---|---|---|---|---|---|---|---|---|
| | | Accuracy | $\Delta_{SP}$ | $\Delta_{EO}$ | Accuracy | $\Delta_{SP}$ | $\Delta_{EO}$ | Accuracy | $\Delta_{SP}$ | $\Delta_{EO}$ |
| **GCN** | 2 Layers | 70.76±0.39 | 17.22±0.95 | 15.43±1.03 | 70.21±0.68 | 8.93±2.25 | 8.50±2.23 | 93.47±0.48 | **11.19±2.33** | 17.75±2.44 |
| | 10 Layers | 68.32±0.58 | **3.09±2.08** | **3.89±0.62** | 68.42±0.49 | **5.66±3.67** | **6.05±3.53** | 91.51±1.42 | **2.86±1.98** | **7.38±1.60** |
| **GraphSAGE** | 2 Layers | 70.16±0.47 | 5.60±2.11 | 3.96±1.87 | 68.43±0.30 | 0.98±0.78 | 2.44±1.74 | 93.62±0.44 | 9.62±3.29 | 16.60±3.31 |
| | 8 layers | 69.74±0.38 | **3.94±1.32** | 3.48±0.77 | 69.13±0.82 | 1.52±1.24 | 2.48±1.20 | 93.52±1.76 | **7.82±3.36** | **11.76±3.97** |
| **APPNP** | 2 Layers | 68.63±1.18 | 17.85±1.63 | 16.39±1.68 | 67.90±0.64 | 9.77±1.26 | 9.53±1.23 | 91.51±0.64 | 13.20±4.21 | 17.30±4.95 |
| | 12 Layers | 68.54±0.15 | **11.09±1.37** | **8.93±1.40** | 68.54±0.09 | **3.09±2.25** | **3.95±0.89** | 92.26±0.92 | **7.39±2.19** | **13.56±2.92** |
| **SGC** | 1-hop | 67.19±0.89 | **16.23±1.32** | 14.71±1.32 | 64.32±2.67 | 7.51±1.17 | 7.36±1.30 | 92.31±1.54 | 10.25±2.31 | 15.44±2.13 |
| | 5-hop | 63.74±5.28 | **5.03±4.28** | **5.69±3.43** | 63.94±6.24 | 7.18±4.75 | 7.28±4.94 | 90.80±0.91 | 14.39±3.10 | 21.10±3.67 |
| **FairGNN** | 2 Layers | 69.81±0.80 | 3.47±1.29 | 1.41±0.86 | 68.98±0.67 | 10.48±5.57 | 9.08±4.93 | 90.60±0.47 | 2.35±1.81 | 2.50±1.45 |
| | 4 Layers | 66.32±0.38 | **0.75±0.72** | **1.29±0.73** | 67.40±0.73 | **2.95±1.47** | **4.84±2.62** | 92.26±1.02 | 3.54±2.06 | 10.71±4.31 |

we extended the attack to multilayer victim models with up to 14 layers. For these experiments, the same selection of victim models as Luo et al. (2024) are used but leaving out the last two Fair models, resulting in a test on: GCN, GraphSAGE, APPNP, SGC, and FairGNN. To increase the amount of layers for FairGNN, some changes had to be made to the underlying model architecture. The use of an adversarial minimax training objective in this model made optimization and convergence more challenging, particularly when additional layers were added. To overcome this issue, a number of changes were made to the underlying GAT model. Instead of the original exponential linear unit (ELU), a Gaussian error linear unit (GELU) activation function was used, which helps to reduce dead neurons in deeper networks (Lee, 2023). Our experiments also revealed that this change helps reduce variability between runs. Furthermore, we added layer normalization, kaiming initialization and residual connections, which all help to stabilize training dynamics and increase convergence (Lee, 2023; Liu et al., 2019). The amount of added layers as found in Table 9 was determined using the highest average fairness over the three datasets. One such distribution of metrics can be found in Figure 3. The complete results of these layer experiments are provided in the Appendix I.1 because of space constraints.

The trends are different for every model and every dataset, but there are some common observations among them. Most notably, we see that every model and dataset combination has a layer number at which the unfairness reaches a low point. This indicates that the model can learn patterns from the training data that lower-layer models cannot capture, inadvertently reducing unfairness. Furthermore, the test accuracy decreases with the amount of layers due to slight oversmoothing. This effect is further discussed in Section 5.3. For most models, a decrease in fairness can be observed at around 10 to 12 extra layers. For SGC, this occurs at a differing hop number depending on the dataset. This could be attributed to the increased influence of the poisoned nodes during message passing in the training stage.

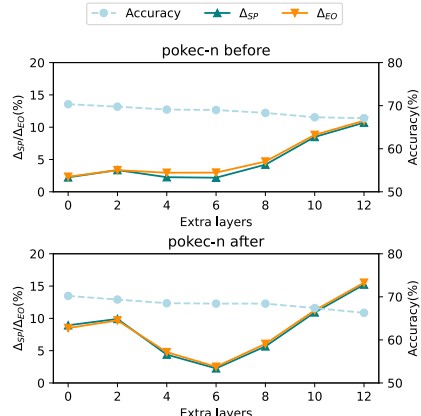

Figure 3: Accuracy and fairness metrics for GCN on the Pokec-n Dataset when adding additional layers.

# 5 Discussion

## 5.1 NIFA as true poisoning attack

We found that NIFA is a combination of a poisoning and evasion attack; it poisons both the training and test set. To clearly distinguish the influence of poisoning the test graph, we have evaluated the first four victim models on NIFA without a poisoned test graph. This can be found in Appendix E. We found that evaluating the poisoned model on a clean test graph significantly reduces the impact of NIFA on the fairness metrics. We hypothesize that the increase of $\Delta_{SP}$ and $\Delta_{EO}$ by utilizing NIFA are caused by the poisoned nodes which are added to the test set, instead of the actual poisoning of the model with a poisoned training set. We observe that our evasion attack increases $\Delta_{SP}$ and $\Delta_{EO}$ over the original NIFA poisoning attack, while

the true poisoning attack shows decreased $\Delta_{SP}$ and $\Delta_{EO}$ values over a model which is trained and evaluated on the clean graph. Meaning, that adding the training on the poisoned graph reduces the model unfairness. These results strengthen our hypothesis about NIFA impacting fairness through the features propagated from the poisoned nodes, instead of the poisoned model itself. It could indicate limited effectiveness of feature-based poisoning attack analogous to NIFA. This is crucial, as it tells us that on one hand, poisoning a model with such an attack has limited effectiveness, while at the same time, existing malicious nodes can have a big impact during inference. We further hypothesize that injecting poison nodes into the training set introduces invariance in the victim model towards the attack. Future work could thus study NIFA as a fair training technique.

## 5.2 Anomalous FairVGNN results on Pokec

As mentioned earlier in Section 3.4, we did not know the exact values of the hyperparameters used for running FairVGNN on the datasets, which required us to do a grid search to get results similar to the authors' original results. Looking at all these results, we noticed many hyperparameter combinations where NIFA actually *improved* the model fairness instead of worsening it and improved or did not affect the accuracy. This was very prominent in the pokec datasets, but did not happen in case of DBLP. Sometimes, all metrics improved on poisoning, but instances can be observed where only the accuracy or one of the fairness metrics improved. The results of all our runs for the Pokec datasets are shown in Table 17 in the Appendix H. We also noticed that these instances of improvement in utility are not rare. Almost half of our runs in pokec-z saw an improvement in at least one metric, and pokec-n was even more extreme, with almost three-quarters of the results having improved metric(s). Perhaps this is not unexpected, since FairVGNN is a fairness-aware model, and it tries to reduce the unfairness by masking the channels highly correlated with sensitive attributes. Additionally, these improvements were within the standard deviation of the clean model results, which may suggest that it might not have *improved*, but at the very least, it manages to be unaffected by NIFA. These results highlight NIFA's sensitivity to victim models' hyperparameters, suggesting that NIFA effectiveness is more variable than the paper indicates. Apart from that, it also shows the potential of limiting sensitive attribute leakage as a defense mechanism against a fairness attack like NIFA. Future work could look at other methods similar to FairVGNN that work on sensitive attribute leakage.

## 5.3 Extensions

**NIFA as evasion attack.** In this study, we evaluate the effectiveness of NIFA as an evasion attack on four different victim models. We found that these models get significantly more impacted on the given fairness metrics compared to the original NIFA framework. We hypothesize that this is due to the invariance learned during training. The victim model inadvertently learns to be fair towards the sensitive attributes if exposed to malicious nodes during training. This does not happen in case of an evasion attack, and we believe this is because the feature-propagation mechanism by which NIFA works, as explained earlier. To further investigate the robustness and effectiveness of this attack, future work can focus on evaluating the application of NIFA as an evasion attack on fairness-oriented models such as FairGNN, FairVGNN and FairSIN.

**Antidote Defense.** The effectiveness of our Antidote Defense was tested by evaluating the extent to which this defense was able to mitigate the effects of the NIFA attack, FA-GNN, FATE, and GFair. As the defender is able to find the best settings for its defence, we used these hyperparameters which are optimal for each individual attack. This shows that just as NIFA can induce unfairness by way of providing neighboring nodes with malicious features, a similar strategy could be used to induce fairness. This is substantiated by the effectiveness of the defense against the FA-GNN attack on the Pokec datasets. It seems to fail on DBLP. However, as mentioned in earlier sections, the problems with the DBLP dataset found in the repository indicates that the results have limited reliability.

**Deeper victim models.** The use of deeper victim models appears to offer some degree of mitigation against unfairness, though the extent of this effect is highly dependent on the specific models and datasets used. This difference is likely due to the properties specific to the graph itself. As for the fairness improvement, since deeper models aggregate features from farther nodes, the impact of the features from malicious nodes is reduced. This shows us that simple interventions without altering the graph structure also have the potential

to limit the effectiveness of an attack such as NIFA. Although, while deeper architectures may help reduce biases, their effectiveness is not uniform across all scenarios. From Table 3 and the results in Appendix I.1, a general decrease in accuracy is observed. This is likely due to the oversmoothing of features, which increases with the amount of layers. The results that validate this claim can be found Appendix I.2.

To fully understand the impact of increasing the number of layers in victim models, a more extensive evaluation is necessary. Future research on this topic can focus on testing a diverse range of victim models in conjunction with additional datasets, which would allow for a more comprehensive assessment of how depth influences fairness across different settings and under varying conditions. Furthermore, future research could use normalization techniques, like layer normalization to decrease the oversmoothing.

### 5.4 What was easy

A significant portion of the reproducibility and replicability experiments were made easy because of the readily available code for their poisoning framework. Additionally, the original authors provided the code for the first four models—GCN, GraphSAGE, APPNP, and SGC—enabling us to efficiently reproduce their results.

### 5.5 What was difficult

**Anomalies:** While the code for some models was available in the repository of the original paper itself, it was not the case for the fairness aware models. This meant that while for some of the models we were simply replicating their results, we had to reproduce the experiments for the fairness-aware models. However, the main difficulty lay with the unavailability of the exact hyperparameters for some models, such as FairVGNN. The authors mention in their paper that they searched some parameters from an array of values, but did not specify which exact values gave them the results they reported. This meant that we had to do a grid search and find the results which were closest to the original results. Furthermore, some discrepancies were found between the documented amount of nodes in DBLP and the ones found in the repository of Luo et al. (2024). Furthermore, we found an overlap between the train, test and validation set. 317 nodes are found in at least two of the splits. 106 of the nodes found in the training set are present in the test set, accounting for approximately 23% of the training set. Using the original dataset, from Hussain et al. (2022), required some pre-processing to be compatible with the code used in this paper. Applying NIFA to this original dataset, yielded significantly different results, which are detailed in Appendix C.

**Environment:** Setting up the environment was not straightforward due to the authors' use of an outdated Python version and obsolete packages, which were incompatible with our computing environment. Additionally, each fairness-aware model required its own environment setup, as they originated from different repositories.

### 5.6 Communication with original authors

Communication with the authors was very good, and they were very responsive. Unfortunately, they did not answer our question about FairVGNN hyperparameters adequately, but apart from that, they were very swift, concise, and co-operative in their responses.

## 6 Conclusion

In our experiments, we were successfully able to replicate and reproduce the authors' results and experiments, some more accurately than others. We can clearly see that NIFA is an effective fairness attack, validating one of the main claims of the original paper. However, we also had some anomalous results in the case of FairVGNN, where NIFA not only seemed to be ineffective, but also worked in the direction opposite to what was intended. Apart from this, we modeled NIFA as an evasion attack, and show that it is an effective method for degrading fairness of GNNs. Using the results from this evasion attack, we hypothesize that utilizing NIFA during training improved model fairness. Lastly, we propose a defense strategy against NIFA, which works to improve fairness of the graph.

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

## A Environmental Impact

To understand the computational feasibility of the NIFA-attack and environmental impacts, we express the CO2-equivalent (Equation 5) of our reproduction experiments.

$$\mathrm{CO_2}e = CI \cdot PUE \cdot P \cdot t \tag{5}$$

Here, $CI$ is carbon intensity in $kg$ $\mathrm{CO_2}e$/kWh, $PUE$ is power usage effectiveness, the ratio that describes how efficiently a data-center uses its power, $P$ is wattage in the used system, and $t$ is the time needed to perform the computations in hours. For the NIFA-attack, we observed an average of 40 seconds needed for performing the attack. These experiments were performed on a HPC-cluster at Amsterdam Science Park. This HPC-cluster runs at a $PUE$ of 1.19 (Dolas, 2022), and a node was used that consumes 0.375 kilowatts of power. Taking the Dutch average of carbon intensity as of 2023; $220g$ $\mathrm{CO_2}e$/kWh (CBS (Statistics Netherlands), 2024), we obtain a CO2-equivalent of:

$$0.22 \cdot 1.19 \cdot 0.375 \cdot \frac{40}{3600} \approx 0.00109 \text{ kg } \mathrm{CO_2}e$$
$$\approx 1.09 \text{ g } \mathrm{CO_2}e.$$

## B Main Attack results

This section contains the main reproducibility results comparing our runs of NIFA on the 7 models and 3 datasets with those obtained by the original authors.

Table 10: Our Accuracy and Fairness performance of NIFA on different victim models compared to the reference found in Luo et al. (2024). The results are reported in percentage (%).

| | | | Pokec-z | | | Pokec-n | | | DBLP | | |
|---|---|---|---|---|---|---|---|---|---|---|---|
| | | | Accuracy | $\Delta_{SP}$ | $\Delta_{EO}$ | Accuracy | $\Delta_{SP}$ | $\Delta_{EO}$ | Accuracy | $\Delta_{SP}$ | $\Delta_{EO}$ |
| Our | GCN | before | 71.50 ± 0.14 | 7.84 ± 1.07 | 5.87 ± 1.04 | 70.34 ± 0.71 | 2.23 ± 1.25 | 2.33 ± 1.33 | 96.58 ± 0.49 | 4.36 ± 1.22 | 2.42 ± 1.03 |
| | | after | 70.76±0.39 | **17.22±0.95** | **15.43±1.03** | 70.21±0.68 | **8.93±2.25** | **8.50±2.23** | 93.47±0.48 | **11.19±2.33** | **17.75±2.44** |
| Ref | | before | 71.22±0.28 | 7.13±1.21 | 5.10±1.28 | 70.92±0.66 | 0.88±0.62 | 2.44±1.37 | 95.88±1.61 | 3.84±0.34 | 1.91±0.75 |
| | | after | 70.50±0.30 | **17.36±1.16** | **15.59±1.08** | 70.12±0.37 | **10.10±2.10** | **9.85±1.97** | 93.37±1.48 | **13.49±2.83** | **20.33±3.82** |
| Our | GraphSAGE | before | 70.70±0.66 | 4.49±1.31 | 3.43±0.95 | 69.14±0.87 | 0.96±0.67 | 1.39±0.71 | 96.23±0.28 | 2.64±1.73 | 2.39±1.03 |
| | | after | 70.16±0.47 | **5.60±2.11** | **3.96±1.87** | 68.43±0.30 | **0.98±0.78** | **2.44±1.74** | 93.62±0.44 | **9.62±3.29** | **16.60±3.31** |
| Ref | | before | 70.79±0.62 | 4.29±0.84 | 3.46±1.12 | 68.77±0.34 | 1.65±1.31 | 1.65±1.41 | 96.58±0.29 | 4.27±1.09 | 2.78±0.91 |
| | | after | 70.05±1.15 | **6.20±1.63** | **4.20±1.77** | 68.43±0.30 | **3.32±1.88** | **3.56±1.91** | 96.46±0.28 | **10.16±2.24** | **16.65±3.30** |
| Our | APPNP | before | 69.88±0.70 | 7.26±0.48 | 5.53±0.52 | 68.33±0.49 | 4.40±0.65 | 4.73±0.65 | 96.58±0.34 | 3.97±1.56 | 2.58±1.17 |
| | | after | 68.63±1.18 | **17.85±1.63** | **16.39±1.68** | 67.90±0.64 | **9.77±1.26** | **9.53±1.23** | 91.51±0.64 | **13.20±4.21** | **17.30±4.95** |
| Ref | | before | 69.79±0.42 | 6.83±1.25 | 5.07±1.26 | 68.73±0.34 | 3.39±0.28 | 3.71±0.28 | 96.58±0.38 | 3.98±1.18 | 2.20±1.08 |
| | | after | 69.12±0.70 | **18.44±1.41** | **16.85±1.50** | 67.90±0.64 | **13.47±3.22** | **13.52±3.56** | 92.46±0.94 | **13.88±3.20** | **20.20±4.25** |
| Our | SGC | before | 68.37±1.30 | 5.54±1.40 | 3.85±1.19 | 66.48±2.35 | 4.64±2.03 | 4.96±1.98 | 96.63±0.38 | 4.70±0.93 | 3.25±0.72 |
| | | after | 67.19±0.89 | **16.23±1.32** | **14.71±1.32** | 64.32±2.67 | **7.51±1.17** | **7.36±1.30** | 92.31±1.54 | **10.25±2.31** | **15.44±2.13** |
| Ref | | before | 69.09±0.99 | 7.28±1.50 | 5.45±1.42 | 66.95±1.69 | 2.74±0.85 | 3.21±0.78 | 96.63±0.48 | 4.70±1.26 | 3.11±1.24 |
| | | after | 67.83±0.70 | **17.65±1.01** | **16.09±1.06** | 66.72±1.21 | **10.59±2.40** | **10.67±2.61** | 93.88±3.37 | **13.88±3.37** | **20.25±4.44** |
| Our | FairGNN | before | 70.27±0.22 | 0.94±0.39 | 2.98±0.48 | 69.66±0.69 | 3.79±1.22 | 5.97±1.58 | 90.70±0.00 | 2.56±0.00 | 0.32±0.00 |
| | | after | 69.81±0.80 | **3.47±1.29** | **1.41±0.86** | 68.98±0.67 | **10.48±5.57** | **9.08±4.93** | 90.60±0.47 | **2.35±1.81** | **2.50±1.45** |
| Ref | | before | 68.75±1.12 | 1.89±0.63 | 1.51±0.47 | 69.41±0.66 | 1.42±0.35 | 2.32±0.57 | 93.12±1.23 | 1.95±0.99 | 3.09±1.81 |
| | | after | 69.38±0.27 | **5.71±2.52** | **4.22±1.89** | 69.67±0.42 | **6.13±4.81** | **6.33±5.77** | 92.56±1.42 | **5.89±2.52** | **10.48±3.82** |
| Our | FairVGNN | before | 64.67±1.63 | 2.76±1.48 | 2.65±0.70 | 65.42±3.44 | 2.96±1.59 | 2.35±1.22 | 90.30±2.22 | 1.42±1.14 | 3.14±2.20 |
| | | after | 63.44±1.10 | **7.01±2.77** | **5.79±3.06** | 64.41±5.24 | **4.01±2.97** | **3.88±3.20** | 86.78±0.72 | **9.83±2.66** | **18.27±7.78** |
| Ref | | before | 68.57±0.45 | 3.79±0.51 | 2.59±0.59 | 67.77±1.23 | 1.90±1.23 | 3.10±1.20 | 95.18±0.54 | 1.90±0.52 | 2.91±1.05 |
| | | after | 67.65±0.38 | **11.01±2.79** | **9.28±2.87** | 65.74±1.42 | **3.51±1.51** | **3.65±1.56** | 91.56±1.13 | **7.96±1.49** | **13.57±2.57** |
| Our | FairSIN | before | 65.80±1.23 | 2.26±2.28 | 2.62±2.10 | 65.25±1.23 | 6.27±0.92 | 7.98±1.01 | 92.81±3.26 | 3.35±1.89 | 11.47±5.46 |
| | | after | 61.70±3.91 | **1.88±0.99** | **2.33±1.99** | 63.64±0.86 | **5.59±1.03** | **5.93±0.91** | 87.84±1.44 | **12.11±1.73** | **25.96±4.78** |
| Ref | | before | 67.33±0.22 | 1.73±1.49 | 2.61±1.44 | 67.18±0.30 | 0.39±0.89 | 2.40±1.02 | 94.72±0.62 | 0.23±0.15 | 0.45±0.16 |
| | | after | 66.55±0.44 | **9.48±2.62** | **10.39±1.06** | 66.20±0.12 | **11.82±0.75** | **14.58±1.22** | 92.46±0.32 | **10.90±2.12** | **23.65±7.77** |

Table 11: Our accuracy and Fairness performance runs of the baselines compared to our NIFA run. The results are reported in percentage (%). The missing values of FATE and G-FairAttack were not obtained due to High memory usage.

| | | Pokec-z | | | Pokec-n | | | DBLP | | |
| | | Accuracy | $\Delta_{SP}$ | $\Delta_{EO}$ | Accuracy | $\Delta_{SP}$ | $\Delta_{EO}$ | Accuracy | $\Delta_{SP}$ | $\Delta_{EO}$ |
|---|---|---|---|---|---|---|---|---|---|---|
| Our | Clean | $71.50 \pm 0.14$ | $7.84 \pm 1.07$ | $5.87 \pm 1.04$ | $70.34 \pm 0.71$ | $2.23 \pm 1.25$ | $2.33 \pm 1.33$ | $96.58 \pm 0.49$ | $4.36 \pm 1.22$ | $2.42 \pm 1.03$ |
| Ref | | $71.22 \pm 0.28$ | $7.13 \pm 1.21$ | $5.10 \pm 1.28$ | $70.92 \pm 0.66$ | $0.88 \pm 0.62$ | $2.44 \pm 1.37$ | $95.88 \pm 1.61$ | $3.84 \pm 0.34$ | $1.91 \pm 0.75$ |
| Our | FA-GNN | $71.72 \pm 0.77$ | $8.33 \pm 2.60$ | $13.23 \pm 5.18$ | $70.19 \pm 0.42$ | $4.00 \pm 2.17$ | $5.79 \pm 1.52$ | $95.43 \pm 00.89$ | $1.56 \pm 0.83$ | $4.78 \pm 2.74$ |
| Ref | | $69.80 \pm 0.48$ | $6.62 \pm 1.21$ | $8.67 \pm 1.28$ | $70.80 \pm 0.97$ | $2.64 \pm 0.76$ | $3.45 \pm 0.54$ | $95.48 \pm 0.48$ | $3.32 \pm 1.65$ | $8.74 \pm 1.23$ |
| Our | FATE | - | - | - | - | - | - | $90.05 \pm 0.26$ | $2.43 \pm 0.45$ | $8.18 \pm 0.73$ |
| Ref | | - | - | - | - | - | - | $94.87 \pm 0.41$ | $3.62 \pm 1.49$ | $2.12 \pm 1.01$ |
| Our | G-FairAttack | - | - | - | - | - | - | $94.02 \pm 0.14$ | $4.6 \pm 1.02$ | $13.86 \pm 1.05$ |
| Ref | | - | - | - | - | - | - | $95.12 \pm 0.38$ | $6.80 \pm 0.59$ | $2.94 \pm 1.10$ |
| Our | NIFA | $70.76 \pm 0.39$ | $\mathbf{17.22 \pm 0.95}$ | $\mathbf{15.43 \pm 1.03}$ | $70.21 \pm 0.68$ | $\mathbf{8.93 \pm 2.25}$ | $\mathbf{8.50 \pm 2.23}$ | $93.47 \pm 0.48$ | $\mathbf{11.19 \pm 2.33}$ | $\mathbf{17.75 \pm 2.44}$ |
| Ref | | $70.50 \pm 0.30$ | $\mathbf{17.36 \pm 1.16}$ | $\mathbf{15.59 \pm 1.08}$ | $70.12 \pm 0.37$ | $\mathbf{10.10 \pm 2.10}$ | $\mathbf{9.85 \pm 1.97}$ | $93.37 \pm 1.48$ | $\mathbf{13.49 \pm 2.83}$ | $\mathbf{20.33 \pm 3.82}$ |

## C  DBLP

As discussed in Section 3.3 and 5.5, some discrepancies were found in the DBLP dataset. Using the dataset obtained as discussed in these sections, results in significantly different fairness results, shown in Table 12.

Table 12: NIFA performance on the original DBLP dataset. The colored values (+/-) indicate the difference between our runs using this DBLP dataset and our runs using the dataset provided by the authors of Luo et al. (2024).

| | | Accuracy | | $\Delta_{SP}$ | | $\Delta_{EO}$ | |
|---|---|---|---|---|---|---|---|
| GCN | before | 95.92 | -0.66 | 0.70 | -3.66 | 3.93 | +1.51 |
| | after | 94.97 | +1.50 | 3.28 | -7.91 | 4.15 | -13.60 |
| GraphSAGE | before | 95.12 | -1.11 | 1.90 | -0.74 | 1.30 | -1.09 |
| | after | 94.54 | +0.92 | 4.52 | -5.10 | 5.94 | -10.66 |
| APPNP | before | 95.69 | -0.89 | 1.69 | -2.28 | 0.78 | -1.80 |
| | after | 94.72 | +3.21 | 6.11 | -7.09 | 8.07 | -9.23 |
| SGC | before | 95.32 | -1.31 | 1.90 | -2.80 | 1.13 | -2.12 |
| | after | 94.49 | +2.18 | 4.90 | -5.35 | 5.87 | -9.57 |
| FairGNN | before | 94.39 | +3.69 | 5.00 | +2.44 | 6.04 | +5.72 |
| | after | 94.67 | +4.07 | 5.05 | +2.70 | 7.19 | +4.69 |

## D   Graph Sparseness and NIFA

In Section 4, differing results were shown by the DBLP dataset, when comparing to the Pokec datasets. We hypothesize that this is due to the sparse nature of DBLP. DBLP has an average degree of 3, compared to 9 of Pokec. In Table 13 we provide a table summarizing fairness metrics before and after poisoning categorized by node degree. We discretized node degree into three approximately equal-sized bins using the 33rd and 67th percentiles as cut-off points, creating groups for low, medium, and high degree evaluation nodes. Consistent with our observations in fairness differences observed between the Pokec datasets and DBLP (Table 4), we see that fairness is impacted more significantly by NIFA poisoning when node degrees are low. Intuitively, injecting nodes to degrade fairness is more effective in sparse neighborhoods, as the relative influence of each injected node increases when the number of existing nodes is small.

Table 13: Change in fairness metrics before and after performing NIFA when comparing different Node degrees.

| | Group | Before $\Delta_{SP}$ | Before $\Delta_{EO}$ | After $\Delta_{SP}$ | After $\Delta_{EO}$ | Change $\Delta_{SP}$ | Change $\Delta_{EO}$ |
|---|---|---|---|---|---|---|---|
| **Pokec-z** | Low (2-7) | 8.91 | 5.39 | 27.10 | 25.17 | ↑↑ | ↑↑ |
| | Medium (8-22) | 15.58 | 13.58 | 23.53 | 21.65 | ↑ | ↑ |
| | High (23+) | 9.34 | 7.72 | 10.23 | 9.02 | ∼ | ↑ |
| | Avg | 9.24 | 7.38 | 17.36 | 15.70 | ↑↑ | ↑↑ |
| **Pokec-n** | Low (2-6) | 8.31 | 8.16 | 10.78 | 12.31 | ↑ | ↑ |
| | Medium (7-19) | 6.05 | 5.80 | 13.55 | 12.10 | ↑↑ | ↑ |
| | High (20+) | 9.34 | 7.57 | 8.09 | 6.75 | ↓ | ↓ |
| | Avg | 0.72 | 2.95 | 8.12 | 7.69 | ↑↑ | ↑ |
| **DBLP** | Low (2-5) | 9.74 | 7.69 | 14.40 | 22.00 | ↑↑ | ↑↑ |
| | Medium (6-10) | 11.24 | 3.21 | 13.49 | 28.09 | ↑ | ↑↑ |
| | High (11+) | 0.94 | 2.63 | 3.70 | 0.00 | ↑ | ↓ |
| | Avg | 4.83 | 2.76 | 12.30 | 17.90 | ↑↑ | ↑↑ |

## E   NIFA as true poisoning attack

Table 14 shows the increase of fairness when training the model on the poisoned graph produced by NIFA. In true poisoning, we evaluate on a clean graph, instead of the poisoned graph as seen in the original NIFA research.

Table 14: The NIFA attack where the test graph remains unpoisoned: the true poisoning attack, compared to the original NIFA poisoning attack, and no attack at all. The **Bold** values indicate an insignificant impact of NIFA on the $\Delta_{SP}$ or $\Delta_{EO}$ compared to no attack.

| | | Pokec-z Accuracy | Pokec-z $\Delta_{SP}$ | Pokec-z $\Delta_{EO}$ | Pokec-n Accuracy | Pokec-n $\Delta_{SP}$ | Pokec-n $\Delta_{EO}$ | DBLP Accuracy | DBLP $\Delta_{SP}$ | DBLP $\Delta_{EO}$ |
|---|---|---|---|---|---|---|---|---|---|---|
| **GCN** | Clean | 71.50±0.14 | 7.84±1.07 | 5.87±1.04 | 70.34±0.71 | 2.23±1.25 | 2.33±1.33 | 96.58±0.49 | 4.36±1.22 | 2.42±1.03 |
| | Poisoning | 70.76±0.39 | 17.22±0.95 | 15.43±1.03 | 70.21±0.68 | 8.93±2.25 | 8.50±2.23 | 93.47±0.48 | 11.19±2.33 | 17.75±2.44 |
| | True Poisoning | 71.26±0.49 | **3.65±0.90** | **2.07±0.70** | 70.55±0.58 | **0.87±0.39** | **3.14±0.77** | 97.19±0.29 | **5.07±1.27** | **2.76±0.93** |
| **GraphSAGE** | Clean | 70.70±0.66 | 4.49±1.31 | 3.43±0.95 | 69.14±0.87 | 0.96±0.67 | 1.39±0.71 | 96.23±0.28 | 2.64±1.73 | 2.39±1.03 |
| | Poisoning | 70.16±0.47 | 5.60±2.11 | 3.96±1.87 | 68.43±0.30 | 0.98±0.78 | 2.44±1.74 | 93.62±0.44 | 9.62±3.29 | 16.60±3.31 |
| | True Poisoning | 70.80±0.34 | **4.35±2.05** | **4.53±1.18** | 68.96±0.80 | **1.31±1.00** | **2.29±1.14** | 96.18±0.62 | **3.47±1.46** | **3.05±1.20** |
| **APPNP** | Clean | 69.88±0.70 | 7.26±0.48 | 5.53±0.52 | 68.33±0.49 | 4.40±0.65 | 4.73±0.65 | 96.58±0.34 | 3.97±1.56 | 2.58±1.17 |
| | Poisoning | 68.63±1.18 | 17.85±1.63 | 16.39±1.68 | 67.90±0.64 | 9.77±1.26 | 9.53±1.23 | 91.51±0.64 | 13.20±4.21 | 17.30±4.95 |
| | True Poisoning | 68.83±1.44 | **6.15±1.56** | **4.73±1.18** | 68.83±0.45 | **4.23±0.38** | **4.57±0.39** | 96.83±0.34 | **2.21±1.25** | **1.57±0.87** |
| **SGC** | Clean | 68.37±1.30 | 5.54±1.40 | 3.85±1.19 | 66.48±2.35 | 4.64±2.03 | 4.96±1.98 | 96.63±0.38 | 4.70±0.93 | 3.25±0.72 |
| | Poisoning | 67.19±0.89 | 16.23±1.32 | 14.71±1.32 | 64.32±2.67 | 7.51±1.17 | 7.36±1.30 | 92.31±1.54 | 10.25±2.31 | 15.44±2.13 |
| | True Poisoning | 66.20±3.63 | **5.64±1.14** | **4.34±0.44** | 67.34±0.63 | **3.90±0.89** | **4.22±0.92** | 96.38±0.41 | **2.44±0.40** | **2.98±1.53** |

# F    Antidote defense on evasion attack

Our Antidote defense showed significant reduction in both $\Delta_{SP}$ and $\Delta_{EO}$ on NIFA. To test the generalizability, we defended against the evasion attack. Defense, 'good', nodes are added before training on the clean graph. After training the victim model is evaluated using a poisoned test graph. Table 15 shows the improvement in fairness after using the Antidote defense on the evasion attack. Using this defense on the evasion attack of GCN on Pokec-n seems to be an outlier, showing higher $\Delta_{SP}$ and $\Delta_{EO}$ than the evasion attack without defense.

Table 15: Accuracy and Fairness metrics comparing the clean model, evasion attack and antidote defense on the evasion attack. The **bold** values indicate a significant decrease in fairness compared to models under the evasion attack with no defense

| | | Pokec-z | | | Pokec-n | | | DBLP | | |
|---|---|---|---|---|---|---|---|---|---|---|
| | | Accuracy | $\Delta_{SP}$ | $\Delta_{EO}$ | Accuracy | $\Delta_{SP}$ | $\Delta_{EO}$ | Accuracy | $\Delta_{SP}$ | $\Delta_{EO}$ |
| **GCN** | Clean | 71.50±0.14 | 7.84±1.07 | 5.87±1.04 | 70.34±0.71 | 2.23±1.25 | 2.33±1.33 | 96.58±0.49 | 4.36±1.22 | 2.42±1.03 |
| | Evasion | 70.80±0.25 | 22.50±1.60 | 20.79±1.68 | 70.21±0.40 | 11.79±3.79 | 11.79±3.79 | 91.76±0.58 | 15.08±2.95 | 22.02±3.30 |
| | Antidote | 71.23±0.35 | **17.01±1.14** | **15.08±1.18** | 69.24±0.84 | 17.59±5.16 | 17.32±4.93 | 88.64±0.72 | 15.26±3.08 | 20.41±2.06 |
| **GraphSAGE** | Clean | 70.70±0.66 | 4.49±1.31 | 3.43±0.95 | 69.14±0.87 | 0.96±0.67 | 1.39±0.71 | 96.23±0.28 | 2.64±1.73 | 2.39±1.03 |
| | Evasion | 69.07±0.53 | 18.97±5.31 | 17.38±5.37 | 67.47±0.50 | 17.81±6.15 | 17.39±6.15 | 90.50±1.42 | 17.00±2.63 | 24.15±2.65 |
| | Antidote | 69.41±0.65 | **4.93±3.40** | **4.34±2.56** | 67.43±0.95 | **12.01±3.40** | **11.73±3.32** | 91.01±2.20 | **7.04±3.57** | **10.52±3.48** |
| **APPNP** | Clean | 69.88±0.70 | 7.26±0.48 | 5.53±0.52 | 68.33±0.49 | 4.40±0.65 | 4.73±0.65 | 96.58±0.34 | 3.97±1.56 | 2.58±1.17 |
| | Evasion | 68.72±0.69 | 21.55±3.59 | 20.11±3.73 | 68.15±0.38 | 12.32±4.43 | 11.96±4.44 | 87.79±2.15 | 19.95±2.06 | 26.30±1.49 |
| | Antidote | 67.55±0.48 | **7.15±1.14** | **5.41±1.12** | 67.91±0.49 | **9.78±1.62** | **9.58±1.69** | 91.11±2.05 | **10.44±4.91** | **15.11±5.64** |
| **SGC** | Clean | 68.37±1.30 | 5.54±1.40 | 3.85±1.19 | 66.48±2.35 | 4.64±2.03 | 4.96±1.98 | 96.63±0.38 | 4.70±0.93 | 3.25±0.72 |
| | Evasion | 67.08±0.89 | 23.07±2.68 | 21.66±2.68 | 66.91±0.33 | 14.66±3.82 | 14.28±3.82 | 86.93±1.78 | 20.64±1.79 | 26.80±1.19 |
| | Antidote | 63.89±1.72 | **4.12±1.53** | **3.21±1.33** | 65.63±0.98 | **9.63±7.77** | **9.43±7.68** | 90.85±1.55 | **12.44±4.70** | **11.96±4.98** |

# G    Masking and Antidote together

After observing the generalizability of the Masking defense in Section 4.3 on other attacks, a combination of both the Antidote and Masking defense was used. Using this combination of defenses, a more generalizable and effective defense could be created. First, using the Masking defense uncertain nodes are masked. Then using the Antidote defense, nodes are added which aim to optimize the fairness of the classification. The results for this ensemble defense are summarized in Table 16. This Table shows that the ensemble defense does not generalize, achieving higher $\Delta_{SP}$ and $\Delta_{EO}$ than the Antidote defense.

Table 16: Accuracy and Fairness metrics comparing the ensemble defense , the Antidote defense and node masking defense against a clean graph and a poisoned graph without the defense. The **Bold** values indicate a significant decrease in $\Delta_{SP}$ or $\Delta_{EO}$ after using the defense.

| | Pokec-z | | | Pokec-n | | |
|---|---|---|---|---|---|---|
| | Accuracy | $\Delta_{SP}$ | $\Delta_{EO}$ | Accuracy | $\Delta_{SP}$ | $\Delta_{EO}$ |
| Clean | 71.50±0.14 | 7.84±1.07 | 5.87±1.04 | 70.34±0.71 | 2.23±1.25 | 2.33±1.33 |
| FA-GNN | 71.72 ± 0.77 | 8.33 ± 2.60 | 13.23 ± 5.18 | 70.19 ± 0.42 | 4.00 ± 2.17 | 5.79 ± 1.52 |
| + Masking | 70.15±0.37 | 9.17±1.25 | 7.18±1.40 | 69.70±0.65 | **1.74±0.82** | **3.11±0.59** |
| + Antidote | 69.83±0.61 | **2.19±1.65** | **2.64±0.72** | 69.35±0.53 | 2.53±0.53 | 3.56±0.98 |
| + Both | 68.82±0.34 | 6.70±4.20 | 9.60±3.36 | 67.97±0.39 | 13.04±2.39 | 13.42±2.40 |

# H   Anomalous FairVGNN results

As discussed in Section 5.2, we found anomalies in the performance of FairVGNN on the Pokec datasets. These results are shown in Table 17, where some combinations of hyperparameters do significantly reduce the influence of NIFA on $\Delta_{SP}$ and $\Delta_{EO}$.

Table 17: FairVGNN results for Pokec datasets. d, g, and c refer to the discriminator, generator, and classifier, respectively. The instances where the attack improved utility or fairness are highlighted in bold

| | Epochs | | | | Pokec-z | | | Pokec-n | | |
|---|---|---|---|---|---|---|---|---|---|---|
| $\epsilon$ | d | g | c | State | Acc | $\Delta_{SP}$ | $\Delta_{EO}$ | Acc | $\Delta_{SP}$ | $\Delta_{EO}$ |
| 1.0 | 5 | 5 | 5 | Before | 67.67±0.84 | 1.34±0.97 | 1.63±0.76 | 63.19±4.93 | 2.55±0.83 | 1.42±1.07 |
| | | | | After | 66.07±2.76 | 3.72±2.29 | 4.39±1.81 | **64.38±3.30** | **2.12±1.21** | 1.76±1.06 |
| 0.1 | 5 | 5 | 5 | Before | 66.52±1.36 | 2.70±1.47 | 2.31±1.41 | 65.23±4.16 | 3.08±1.40 | 2.48±1.23 |
| | | | | After | 66.27±1.30 | 4.44±1.64 | 3.65±2.52 | **67.01±1.07** | **2.87±1.19** | 2.88±0.93 |
| 0.01 | 5 | 5 | 5 | Before | 62.91±2.49 | 3.50±3.85 | 2.43±3.20 | 63.33±4.41 | 4.00±2.28 | 3.73±2.22 |
| | | | | After | **63.38±1.88** | **1.30±0.64** | **1.82±0.44** | **66.11±1.39** | **2.92±2.20** | **2.82±1.68** |
| 1.0 | 5 | 5 | 10 | Before | 63.17±2.19 | 3.86±1.74 | 2.85±1.64 | 65.42±3.44 | 2.96±1.59 | 2.35±1.22 |
| | | | | After | **64.86±2.51** | **3.00±2.64** | **2.77±2.10** | 64.41±5.24 | 4.01±2.97 | 3.88±3.20 |
| 0.1 | 5 | 5 | 10 | Before | 66.07±1.33 | 3.64±2.37 | 3.11±1.60 | 64.04±4.52 | 5.81±3.93 | 4.04±2.75 |
| | | | | After | **66.17±0.64** | **3.18±3.65** | 3.62±2.67 | **65.71±1.84** | **3.97±2.50** | **2.98±3.72** |
| 0.01 | 5 | 5 | 10 | Before | 59.59±1.88 | 2.32±1.69 | 1.22±0.52 | 65.48±1.98 | 3.09±1.91 | 3.85±1.53 |
| | | | | After | **60.63±0.59** | 5.62±2.15 | 3.57±2.11 | 61.61±6.56 | 4.29±2.60 | **2.40±2.64** |
| 1.0 | 5 | 10 | 5 | Before | 67.15±2.65 | 1.86±1.55 | 1.74±1.13 | 65.37±2.69 | 4.04±1.81 | 3.24±1.39 |
| | | | | After | **67.17±1.23** | 4.83±3.12 | 4.80±1.94 | 64.85±3.13 | **3.29±1.42** | **2.60±0.65** |
| 0.1 | 5 | 10 | 5 | Before | 67.34±1.43 | 2.24±1.40 | 2.09±1.69 | 65.38±2.54 | 2.92±1.79 | 2.16±1.36 |
| | | | | After | 66.35±1.00 | 5.13±4.52 | 4.98±4.15 | **65.99±2.30** | 9.36±3.97 | 7.40±4.29 |
| 0.01 | 5 | 10 | 5 | Before | 60.72±2.31 | 2.99±2.50 | 2.17±1.26 | 66.15±1.48 | 3.69±2.19 | 4.05±1.82 |
| | | | | After | **62.17±2.40** | **2.57±2.19** | 2.17±2.19 | 60.86±6.12 | **2.44±0.85** | **2.94±1.81** |
| 1.0 | 10 | 5 | 5 | Before | 67.00±1.64 | 3.33±2.88 | 2.11±2.21 | 66.53±1.90 | 3.01±2.05 | 2.13±1.33 |
| | | | | After | 65.63±1.84 | 5.53±4.94 | 5.45±2.97 | **67.20±0.81** | **2.66±1.87** | 3.18±1.78 |
| 0.1 | 10 | 5 | 5 | Before | 64.86±2.79 | 10.71±7.71 | 7.18±5.00 | 67.16±1.46 | 3.71±3.07 | 3.21±2.24 |
| | | | | After | **65.95±1.88** | **5.76±4.37** | **4.75±4.64** | 61.96±3.47 | 10.52±5.43 | 7.79±3.32 |
| 0.01 | 10 | 5 | 5 | Before | 60.94±3.67 | 0.91±1.32 | 1.91±1.70 | 58.67±5.65 | 12.18±9.29 | 9.23±6.80 |
| | | | | After | 59.52±2.63 | 3.88±4.67 | 2.67±3.50 | 56.89±6.90 | **5.22±6.57** | **6.06±8.54** |
| 1.0 | 5 | 10 | 10 | Before | 67.79±0.83 | 1.99±0.97 | 3.30±0.95 | 65.79±2.38 | 2.87±1.71 | 1.73±1.40 |
| | | | | After | 64.43±1.88 | 3.49±2.65 | **2.80±2.36** | 64.01±3.02 | 6.04±4.12 | 4.15±2.59 |
| 0.1 | 5 | 10 | 10 | Before | 66.48±0.98 | 1.38±0.99 | 1.33±1.22 | 68.65±0.56 | 4.77±3.04 | 5.51±2.84 |
| | | | | After | 65.64±0.68 | 4.18±2.13 | 3.93±2.06 | 66.99±1.77 | **3.76±2.54** | **3.62±3.02** |
| 0.01 | 5 | 10 | 10 | Before | 60.58±1.14 | 2.48±1.52 | 1.12±1.08 | 65.16±1.14 | 2.51±1.27 | 3.11±1.49 |
| | | | | After | **61.34±1.92** | **2.82±1.44** | **1.18±0.49** | 62.57±5.21 | 3.50±2.30 | **2.73±2.88** |
| 1.0 | 10 | 5 | 10 | Before | 66.45±2.52 | 3.49±1.24 | 3.36±1.48 | 63.92±2.34 | 3.00±1.22 | 3.28±1.51 |
| | | | | After | **66.75±0.54** | **2.91±2.94** | 3.79±2.76 | **64.27±4.66** | 4.82±1.12 | 3.69±1.80 |
| 0.1 | 10 | 5 | 10 | Before | 65.83±0.89 | 2.47±1.65 | 1.95±1.11 | 67.50±1.08 | 3.07±2.82 | 2.42±2.74 |
| | | | | After | 65.58±1.00 | 2.48±2.08 | 2.31±1.15 | 65.15±3.19 | 4.91±2.10 | 3.95±1.79 |
| 0.01 | 10 | 5 | 10 | Before | 61.50±3.47 | 2.50±1.62 | 1.31±0.80 | 66.41±0.88 | 5.67±1.10 | 5.96±1.72 |
| | | | | After | **62.37±2.79** | 2.52±2.50 | 2.13±1.83 | 63.57±2.53 | **4.48±2.93** | **3.56±2.61** |
| 1.0 | 10 | 10 | 5 | Before | 64.67±1.63 | 2.76±1.48 | 2.65±0.69 | 66.49±3.11 | 4.06±2.88 | 3.47±3.06 |
| | | | | After | 63.44±1.10 | 7.01±2.77 | 5.79±3.06 | 65.17±1.98 | 5.09±4.28 | 5.05±5.34 |
| 0.1 | 10 | 10 | 5 | Before | 67.13±0.72 | 7.07±6.02 | 6.98±5.23 | 64.75±6.04 | 7.85±4.12 | 4.59±2.28 |
| | | | | After | 65.22±1.38 | 10.52±10.87 | 8.07±8.02 | **65.44±3.63** | **1.90±1.61** | **1.69±1.39** |
| 0.01 | 10 | 10 | 5 | Before | 58.73±3.87 | 2.35±1.54 | 1.07±0.63 | 60.30±6.46 | 8.17±4.31 | 6.03±3.87 |
| | | | | After | 57.89±2.90 | 8.01±5.87 | 6.66±6.05 | **62.19±2.14** | **4.60±3.77** | **2.56±2.05** |
| 1.0 | 10 | 10 | 10 | Before | 67.65±0.55 | 1.82±0.94 | 1.65±0.99 | 65.17±2.94 | 2.33±2.23 | 2.36±1.64 |
| | | | | After | 65.42±1.28 | 2.44±1.95 | 2.21±2.08 | **66.28±1.46** | 3.56±3.03 | 3.07±2.35 |
| 0.1 | 10 | 10 | 10 | Before | - | - | - | 65.41±3.74 | 3.09±1.49 | 2.04±1.00 |
| | | | | After | - | - | - | **66.95±0.88** | 3.71±1.50 | 4.21±1.42 |
| 0.01 | 10 | 10 | 10 | Before | - | - | - | 66.24±0.59 | 3.63±2.08 | 4.36±2.10 |
| | | | | After | - | - | - | 59.85±5.47 | 4.75±2.62 | **2.48±0.93** |

# I Increasing Model size

## I.1 Main Results

This section contains the complete results for the layer experiments found in Section 4.3 of our research. The plots below indicate the choice of extra layers per model. 0 extra layers denotes the performance as seen in Table 4.

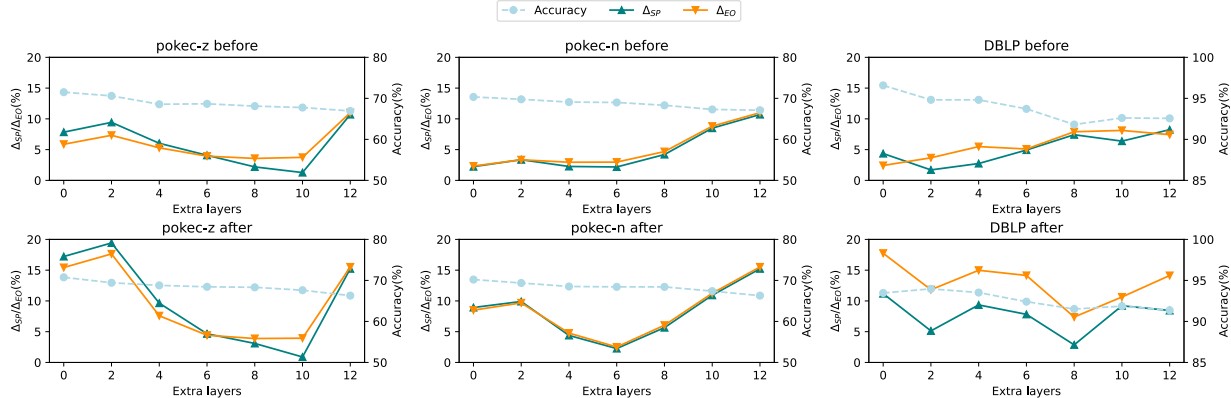

Figure 4: Impact of increasing the number of layers in the GCN model on accuracy, statistical parity, and equal opportunity metrics.

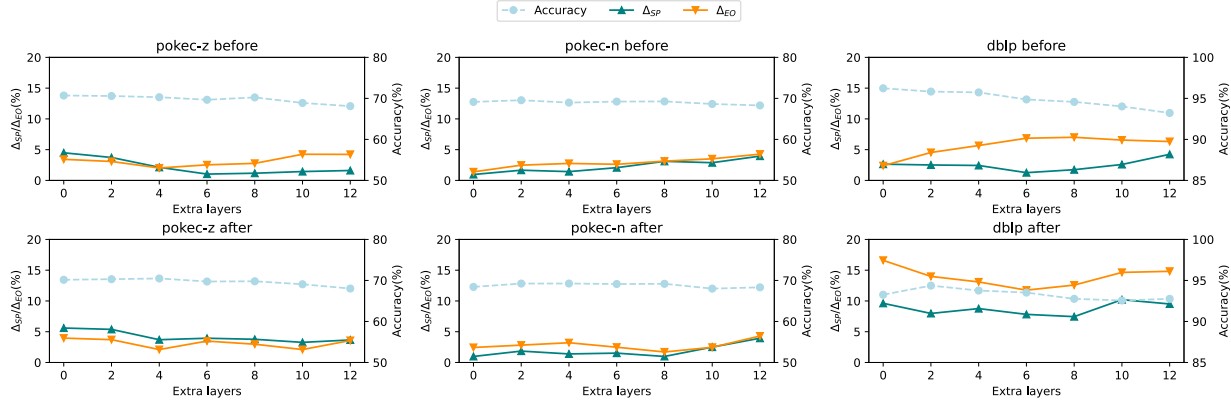

Figure 5: Impact of increasing the number of layers in the GraphSAGE model on the accuracy, Statistical Parity and Equal opportunity metrics.

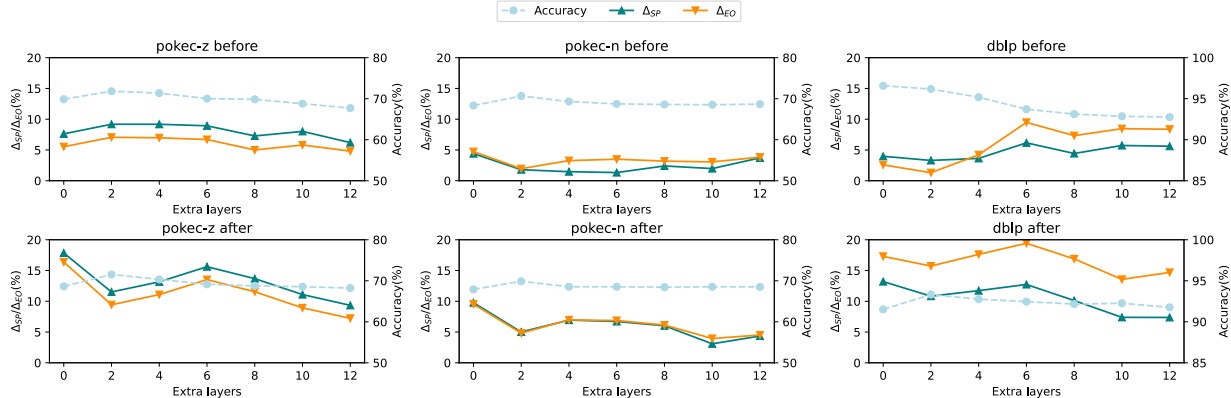

Figure 6: Impact of increasing the number of layers in the APPNP model on the accuracy, Statistical Parity and Equal opportunity metrics.

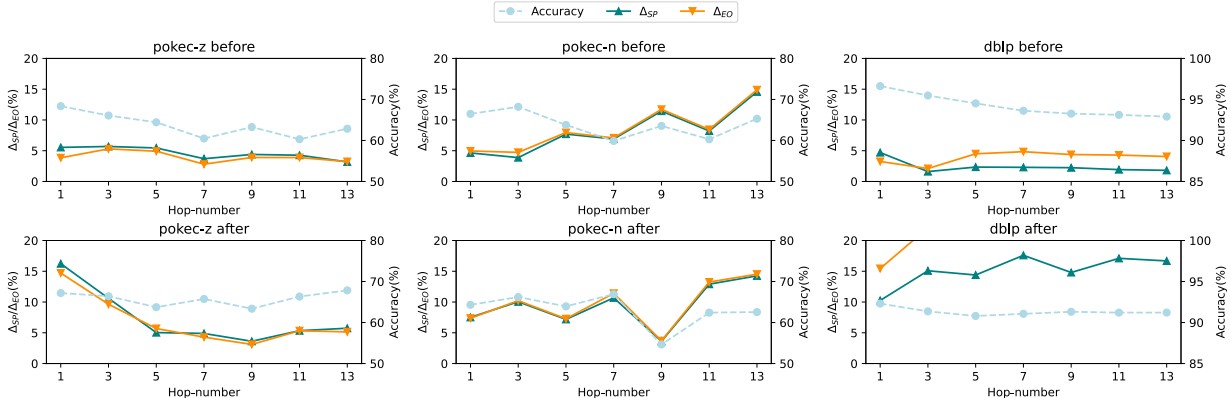

Figure 7: Impact of increasing the number of layers in the SGC model on the accuracy, Statistical Parity and Equal opportunity metrics.

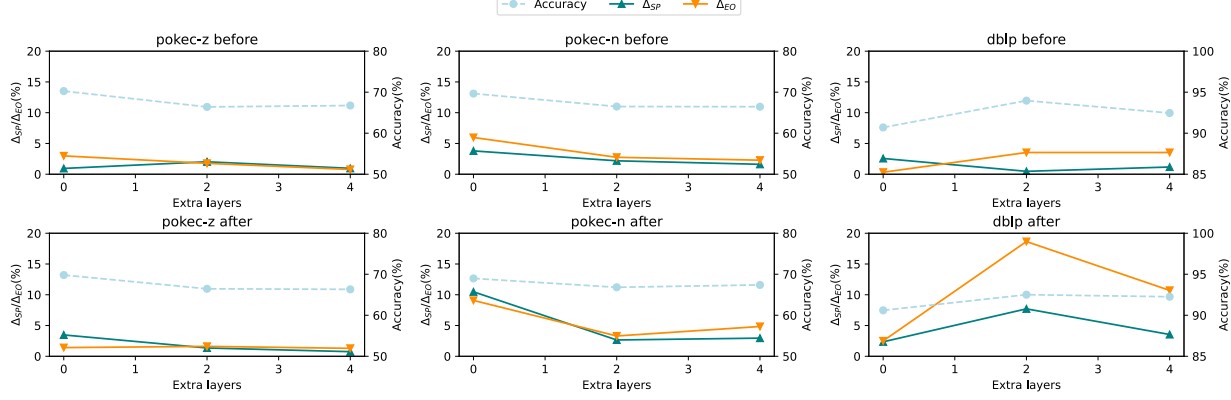

Figure 8: Impact of increasing the number of layers in the FairGNN model on the accuracy, Statistical Parity and Equal opportunity metrics.

## I.2 Oversmoothing

We found that adding layers causes oversmoothing of the features in the penultimate layer, before the last activation function and classification layer. These results are visualized in Figures 9a, 9b, and 10, where the standard deviation between the output features of this penultimate layer is plotted per added layers, averaged over the test set. SGC is left out in these results, as no layers are added to this model. The oversmoothing by the victim models likely causes the decrease in accuracy, $\Delta_{SP}$, and $\Delta_{EO}$. In particular this decrease in unfairness, indicates a tradeoff between utility and fairness, where decreasing the influence of individual neighbors decreases utility, but increases fairness.

The GCN and APPNP models follow a consistent pattern, where the standard deviation of the output features decreases with the amount of added layers. However, on the DBLP dataset, this standard deviation seems to increase with the amount of layers. This might be due to the sparseness of the graph compared to the Pokec datasets, as shown in Appendix D. Increasing the amount of layers increases the amount of aggregated neighbors, but this affect is not as significant on DBLP as on the Pokec datasets.

GraphSAGE shows increased standard deviation of output features when adding more layers. This is expected considering its architecture, where at each layer it considers a fixed neighborhood from which it aggregates features. This means that the features do not get oversmoothed, when adding more layers.

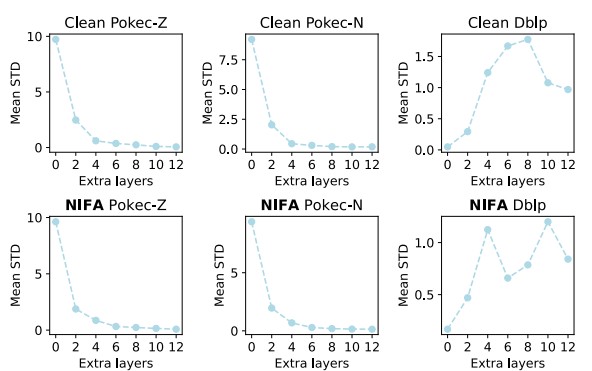

(a) Mean standard deviation of the penultimate layer when adding more layers to the **GCN** victim model.

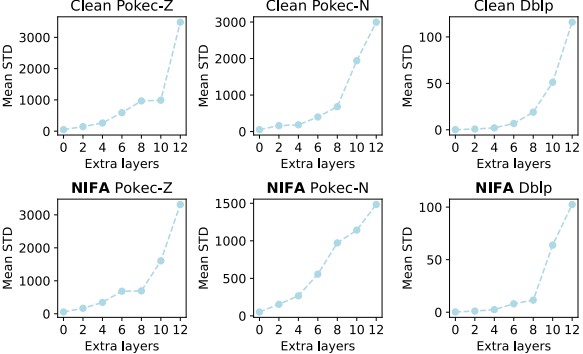

(b) Mean standard deviation of the penultimate layer when adding more layers to the **GraphSAGE** victim model.

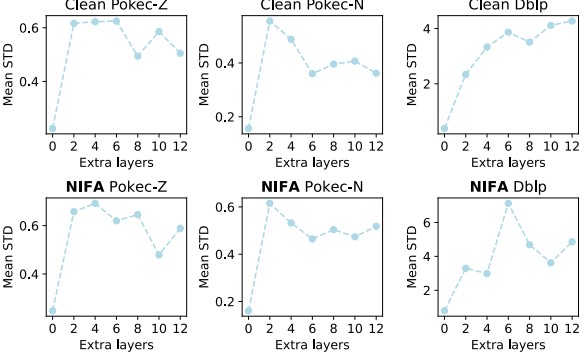

Figure 10: Mean standard deviation of the penultimate layer when adding more layers to the **APPNP** victim model.

