# OpenReview forum: "[RE] Are Your Models Still Fair? Fairness Attacks on Graph Neural Networks via Node Injections"
_TMLR — Rejected by TMLR_

### Review · Reviewer_Pyhr · 2025-03-24

**Summary Of Contributions:**

This paper presents a reproducibility study of Node Injection based Fairness Attack (NIFA), and validates the findings in the original paper. The paper also demonstrates the effectiveness of NIFA as an inference-time evasion attack, where the training graph is not poisoned. Additionally, a potential defense against NIFA based on injecting "good" nodes into the training graph is proposed, and its effectiveness in countering NIFA is demonstrated.

**Audience:**

No

**Broader Impact Concerns:**

Sufficiently addressed

**Claims And Evidence:**

Yes

**Requested Changes:**

Critical for acceptance:
+ Add more details on why the reproducibility study is needed. In particular, clearly state the differences between the experiments in the original paper and the reproducibility study presented in this paper. Additional questions: (1) Are there missing details in the original paper and released code base that make it difficult to reproduce the results? (2) Are there any specific concerns about NIFA methodology that make it unlikely to generalize to some datasets? If so, can the deficiencies of NIFA be brought out (and described if they are alleviated) in this study?
+ Explicitly state the new insights gained from this reproducibility study that were not part of the original paper, and why these new insights would be useful for the community.
+ Add details on the additional computational complexity introduced by the proposed "antidote" defense. Since new nodes are added to the graph, it is expected that training and inference will now be slower.

Would strengthen the paper:
+ It would be good to evaluate the proposed "antidote" defense on other fairness attacks to demonstrate its generalizability. Since this can be seen as the "inverse" of NIFA, it may be specialized to defend only against NIFA.
+ It is interesting that the injection of "good" nodes during training actually decreased the clean accuracy (Table 7). Since injecting "good" nodes would increase homophily, I would expect it to help in the node classification setting. Do the authors have any insights into this?

**Strengths And Weaknesses:**

Strengths:
+ The paper is clearly written, and all hyperparameters used in the experiments are clearly stated. Source code is also released. So, all results presented in the paper can be reproduced with minimal effort.
+ The extension of NIFA to an evasion attack demonstrates its effectiveness in more realistic scenarios that do not require access to the training data.
+ The proposed defense based on injecting "good" nodes into the graph is interesting, and the results show it can be an effective defense against NIFA.

Weaknesses:
+ The need for the reproducibility study of NIFA is not well motivated, especially since the authors of NIFA have released their code.
+ It is unclear what additional insights are gained from the reproducibility study.
+ The argument for using larger model depths as a means of countering NIFA is not very convincing. In addition to exponentially increasing computational complexity, the results on fairness degradation (and even clean accuracy degradation) are mixed.

---

> ### Author Response · Authors · 2025-04-01
> **Author response to requested changes and Weaknesses**
>
> Thank you for your sharp and detailed insight on our research.
> Below we have added responses to your comments and possible additions to our paper.
>
>
> Requested Changes:
> Critical for acceptance:
> >1.Add more details on why the reproducibility study is needed. In particular, clearly state the differences between the experiments in the original paper and the reproducibility study presented in this paper. Additional questions: (1) Are there missing details in the original paper and released code base that make it difficult to reproduce the results? (2) Are there any specific concerns about NIFA methodology that make it unlikely to generalize to some datasets? If so, can the deficiencies of NIFA be brought out (and described if they are alleviated) in this study?
>
> Response: We have added additional text discussing these concerns in section 2. (1) yes, the original paper was missing important code and hyperparameters needed to gain similar results. We have documented both. Furthermore, discrepancies have been found regarding the DBLP dataset (Section 5.5), and FairVGNN results. (2) The DBLP dataset showed that the attack did not seem to generalize as well as the authors proposed. Furthermore, we show that the added GAT model does generalize to our evasion version of NIFA, but does not effectively reduce fairness when using the original poisoning version (Sections 4.1 & 4.3).
>
>
> >2. Explicitly state the new insights gained from this reproducibility study that were not part of the original paper, and why these >new insights would be useful for the community.
>
> Response: We have added additional text discussion these, in Section 5.1, 5.2, and 5.3. In particular, the inner workings of NIFA are further explained, reasoning about the effectiveness of the evasion version of NIFA, compared to the original poisoning attack.
>
>
> >3. Add details on the additional computational complexity introduced by the proposed "antidote" defense. Since new nodes are added to the graph, it is expected that training and inference will now be slower.
>
> Response: To illustrate the additional computational overhead introduced by our defense, we have included a theoretical explanation of inference time before and after our defense, and the additional computation introduced by feature optimisation. This explanation is provided in Section 4.3 - Antidote Defense.
>
>
>
>
> Weaknesses:
> >The argument for using larger model depths as a means of countering NIFA is not very convincing. In addition to exponentially increasing computational complexity, the results on fairness degradation (and even clean accuracy degradation) are mixed.
>
> Response: We agree that in the older state, the reasoning and importance was lacking. We added further reasoning about the results, including an explanation about the decrease of accuracy, Delta EO, and Delta SP when increasing layers. We found that this decrease correlates positively with the decrease in standard deviation of the features of the penultimate layer (pre activation, before the last classification layer). These can all be found in Section 4.3 Deeper victim models, and 5.3 Deeper victim models.
>
> > It would be good to evaluate the proposed "antidote" defense on other fairness attacks to demonstrate its generalizability. Since this can be seen as the "inverse" of NIFA, it may be specialized to defend only against NIFA.
>
> Response: We have evaluated our defense on other attack methods (FA-GNN, FATE, GFAIR) which are included in section 4.3 table 8.

---

> > ### Comment · Reviewer_Pyhr · 2025-04-15
> > **Response to rebuttal**
> >
> > Thanks to the authors for the responses and revisions. My concerns regarding the generalization of the antidote defense and the use of deeper models have been addressed. However, after reading through the revised paper, it is still difficult for me to understand the primary motivation for conducting the reproducibility study, and how the experiments vary from those in the original paper. Some additional questions on this front:
> >
> > - Did the authors try running the source code provided by the original paper? If so, what were the observations, and were there any discrepancies (apart from the distribution change in DBLP)?
> > - What are the differences between the original experiments and the ones in the reproducibility study? Are the same experiments repeated with a different set of hyperparameters? Is the software environment different (using different versions of packages, and/or entirely different packages)?
> > - Did the authors get insights into why NIFA is highly sensitive to hyperparameters, or how the optimal hyperparameters can be selected from the experiments?

---

> > > ### Author Response · Authors · 2025-04-15
> > > **Response to rebuttal**
> > >
> > > Thank you for your comments. We have changed our pdf slightly to accomodate your changes.
> > >
> > > >Did the authors try running the source code provided by the original paper? If so, what were the observations, and were there any discrepancies (apart from the distribution change in DBLP)?
> > >
> > > Response: Yes, we ran the source code provided by the original paper. However, since the code for fairness-aware models was not included in the repository of the paper, we had to refer to the original papers for the attacks. This is explained in Section 2.
> > >
> > > >What are the differences between the original experiments and the ones in the reproducibility study? Are the same experiments repeated with a different set of hyperparameters? Is the software environment different (using different versions of packages, and/or entirely different packages)?
> > >
> > > Response: The original experiments mainly consisted of obtaining a baseline, poisoning the dataset, training on the poisoned dataset, and evaluating on the poisoned data. For reproducing the results, we run the same experiment on the same hyperparameters. Our extension experiments are different. Our evasion attack switches the order of poisoning the dataset and training. We train on the clean dataset, poison the dataset, and evaluate the dataset. In our results for true poisoning attack, the first 3 steps are the same as the original experiments, but the evaluation is done on a clean graph. In Antidote defense, we defend our graph after attacking with NIFA. Additionally, for reproducing the results of FairVGNN, we do a hyperparameter search because the exact values are not mentioned by the authors.
> > > The hyperparameters (or the set of hyperparameters) we use are the same as the ones in the original paper.
> > > We use a similar software environment because we have the same packages, but we use different versions of the packages. Some information to the pdf was added to explain this. The versions of the packages can be found in our repository.
> > >
> > > >Did the authors get insights into why NIFA is highly sensitive to hyperparameters, or how the optimal hyperparameters can be selected from the experiments?
> > >
> > > Response: We had a sentence in our paper which was quite ambiguous, this has been changed in the pdf. NIFA is not exceptionally sensitive to its own hyperparameters, but the victim models can change the effectiveness of NIFA when changing the victim models’ hyperparameters. For FairVGNN we saw that NIFA is quite ineffective when using specific hyperparameters. This is more clearly explained in Section 5.2.

---

### Review · Reviewer_8vZT · 2025-03-27

**Summary Of Contributions:**

- The paper identifies limitations in existing GNN fairness attacks, particularly in terms of realism and stealthiness. To address this, it proposes NIFA, a novel fairness attack based on node injection, which perturbs fairness without modifying existing nodes or edges.
- Through extensive experiments on both standard GNNs (e.g., GCN, GraphSAGE, APPNP) and fairness-aware models (e.g., FairGNN, FairSIN), the study demonstrates that these models remain vulnerable to fairness degradation under the proposed attack.
- The results reveal that, unlike traditional utility-focused attacks, NIFA can selectively degrade fairness while preserving predictive performance, introducing a new threat model for fairness robustness in GNNs.

**Audience:**

Yes

**Broader Impact Concerns:**

No concerns on broader impact.

**Claims And Evidence:**

Yes

**Requested Changes:**

1. In Section 5.2 and Appendix E, NIFA often improves the fairness of FairVGNN - particularly on Pokec - contrary to its intended purpose. This phenomenon should be more thoroughly investigated and explained, particularly in light of FairVGNN’s design to suppress sensitive attribute leakage.

2. The proposed Antidote Defense achieves empirical results but lacks a clear explanation and justification.  It would strengthen the contribution to explain why fairness-aware feature optimization via node injection leads to such consistent mitigation. Furthermore, a discussion on whether this defense generalizes to other fairness attacks (e.g., FA-GNN, FATE) would broaden its relevance.

3. Section 4.3 presents experiments on deeper GNN architectures. However, the discussion remains descriptive. A more rigorous analysis of why deeper models mitigate NIFA - possibly  related to over-smoothing, feature propagation distance, etc. - would greatly improve this section..

4. The current attack formulation assumes access to surrogate GNN gradients for adversarial feature optimization. While common in literature, this assumption may not reflect practical gray-box settings where gradients are typically inaccessible. Rather than relabeling the threat model, I suggest acknowledging this limitation and discussing how NIFA could be adapted for more restricted settings (e.g., gradient-free optimization).

5. Section 3.3 notes that only a subset of labeled nodes in DBLP is used, citing missing sensitive attributes and differences from the source dataset in Hussain et al. (2022). To ensure reproducibility, it would be helpful to clarify whether the dataset version matches that used in Luo et al. (2024), and what specific preprocessing was applied.

6. The evaluation relies on both $\Delta$SP and $\Delta$EO. As I know, those are binary-sensitive-attribute metrics. However, datasets such as DBLP may contain multi-class structures or sensitive attribute distributions. The authors need to clarify whether these metrics are sufficient and discuss the need for multi-class fairness metrics.

7. Luo et al. show that increasing the removal of uncertain nodes (via $\eta$) can improve fairness with minimal utility loss. Accoddingly, it seems that removing uncertain nodes  can reduce the fairness metrics, but I wonder if the author can discuss it further in the context of previous work, although the attack type is different.

8. The statement _“evaluating the poisoned model on a clean train graph significantly reduces the impact of NIFA”_ is potentially misleading. It seems that “train” was intended to mean “test” in this context. Please clarify the intended meaning.

**Strengths And Weaknesses:**

**Strenghts**

1. This paper introduces NIFA, a node injection-based attack targeting fairness in GNNs, and presents a realistic and stealthy threat model that is more applicable to real-world scenarios compared to prior fairness attacks.
2. It conducts comprehensive empirical evaluations across various GNN architectures, including fairness-aware models such as FairGNN and FairSIN, demonstrating that NIFA can effectively induce fairness degradation.
3. The proposed method integrates uncertainty-based node selection, a homophily-increase principle, and feature optimization into a unified attack framework, balancing attack efficiency and practical feasibility.

**Weaknesses**

1. The authors redefine NIFA as an evasion attack, yet do not provide sufficient theoretical justification for how this diverges from its original categorization as a poisoning attack. Since the two attack types differ fundamentally in terms of when and how the injected nodes influence the model (training-time vs. inference-time effects), a more formal and explicit clarification would help avoid conceptual confusion.

2. In Section 5.1 and Appendix C, the authors report that a “true poisoning attack” (i.e., poisoning only the training graph) leads to a _reduction_ in fairness degradation relative to the full NIFA attack. This directly challenges the claim that NIFA consistently undermines fairness and may leave readers unclear about the actual mechanism through wich NIFA affects model fairness.

3. While the proposed Antidote Defense mirrors the NIFA optimization pipeline in the opposite direction, it remains unclear why such a defense should generalize beyond NIFA to other fairness attacks (e.g., FA-GNN, FATE, G-FairAttack). Either theoretical reasoning or additional experiments would strengthen the case for the defense's general applicability.

4. In several cases — particularly on the Pokec dataset — the application of NIFA appears to _improve_ fairness metrics ($\Delta$SP and $\Delta$EO). If the attack does not consistently degrade fairness, the stealthiness and effectiveness of NIFA as a fairness attack become less convincing. A more detailed discussion of such failure or edge cases would help clarify the scope and limitations of the attack.

5. On the DBLP dataset, the observed accuracy drops from 3% to over 5%, depending on the model and attack mode (e.g., evasion). These changes are significant and suggest that the claim of NIFA having no utility impact may be overstated. It would be more accurate to qualify the extent of the utility-preserving property.

---

> ### Author Response · Authors · 2025-04-01
> **Author response to Requested Changes (1/2)**
>
> Thank you for your thoughtful feedback. We appreciated having a fresh outside view on the research.
> Below are answers and our additions to the comments. Please also check out our improved pdf.
>
>
> > Requested Changes:
> > 1. In Section 5.2 and Appendix E, NIFA often improves the fairness of FairVGNN - particularly on Pokec - contrary > to its intended purpose. This phenomenon should be more thoroughly investigated and explained, particularly in light > of FairVGNN’s design to suppress sensitive attribute leakage.
>
> Response: The reasoning why this happens is twofold:
> In 5.1 we added text improving the explanation why NIFA adds invariance to the victim model to itself. Adding the training on the poisoned graph reduces the model unfairness. These results strengthen our hypothesis about NIFA impacting fairness through the features propagated from the poisoned nodes, instead of the poisoned model itself. It could indicate limited effectiveness of feature-based poisoning attacks analogous to NIFA. These points can be deduced from the NIFA as ‘true poisoning attack’, original NIFA, and NIFA as evasion attack.
> And from in 5.2, we have added:
> These results highlight the sensitivity of NIFA to hyperparameters, suggesting that its effectiveness is more variable than the paper indicates. Apart from that, it also shows the potential of limiting sensitive attribute leakage as a defense mechanism against a fairness attack like NIFA.
>
>
> >2. The proposed Antidote Defense achieves empirical results but lacks a clear explanation and justification. It would strengthen the contribution to explain why fairness-aware feature optimization via node injection leads to such consistent mitigation. Furthermore, a discussion on whether this defense generalizes to other fairness attacks (e.g., FA-GNN, FATE) would broaden its relevance.
>
> Response: We evaluated our defense on other attacks as well and have added the results to the paper in Table 8. We also added an explanation about why this works - which is analogous to how the attack itself works (Section 5.3, Antidote Defense). The attack optimizes the features such that when propagated to other nodes, they induce unfairness. Similarly, by minimizing a different objective, we optimize the features such that by feature propagation, they reduce the unfairness (or induce fairness) during classification. Do keep in mind that we are still doing a hyperparameter search to find their influence on the effectiveness of Antidote defense.
>
>
> >3. Section 4.3 presents experiments on deeper GNN architectures. However, the discussion remains descriptive. A more rigorous analysis of why deeper models mitigate NIFA - possibly related to over-smoothing, feature propagation distance, etc. - would greatly improve this section.
>
> Response: We have added an explicit explanation for deeper models mitigating NIFA, which we believe is related to the feature propagation distance (Section 5.3, Deeper Victim Models). Additionally, to understand the degradation in performance after adding more layers, we have added results of the variance in features before the classification layer of the model (Figure 10). These experiments indicate that oversmoothing is likely the cause for degradation in performance at higher depths, as the reducing of variance in features highly correlates with reducing unfairness.
>
>
> > 4. The current attack formulation assumes access to surrogate GNN gradients for adversarial feature optimization. While common in literature, this assumption may not reflect practical gray-box settings where gradients are typically inaccessible. Rather than relabeling the threat model, I suggest acknowledging this limitation and discussing how NIFA could be adapted for more restricted settings (e.g., gradient-free optimization).
>
> Response: This assumption of accessing the gradients of the surrogate model is quite a realistic setting, as we assume a setting where the attacker has access to its own model (the surrogate model: always a GCN model for NIFA) which is different from the victim model (Like GCN, GraphSAGE, FairGNN, etc). The only access that the attacker has, are the labels, features, and sensitive attributes.
> These assumptions mean that NIFA is actually a gradient-free optimization method, with respect to the victim model. The gradients are obtained from the attacker's surrogate model.

---

> ### Author Response · Authors · 2025-04-01
> **Author response to Requested Changes (2/2)**
>
> >5. Section 3.3 notes that only a subset of labeled nodes in DBLP is used, citing missing sensitive attributes and differences from >the source dataset in Hussain et al. (2022). To ensure reproducibility, it would be helpful to clarify whether the dataset version >matches that used in Luo et al. (2024), and what specific preprocessing was applied.
>
> Response: In Section 3.3 (Datasets), we have added an explicit mention at the end that the DBLP dataset we use for our experiments is the one found in the original repository. Additional clarification of the discrepancies found between the original DBLP dataset and the dataset supplied in the NIFA authors’ repository has been added to Section 5.5.
>
>
> >6. The evaluation relies on both ΔSP and ΔEO. As I know, those are binary-sensitive-attribute metrics. However, datasets such >as DBLP may contain multi-class structures or sensitive attribute distributions. The authors need to clarify whether these >metrics are sufficient and discuss the need for multi-class fairness metrics.
>
> Response: We agree that generalising the attack and defense to multi-class sensitive labels can reveal additional insights, and added an acknowledgement and clarification to why we use binary sensitive attributes to the introduction and section on fairness metrics (Section 5.3). The components of the graphs we use in our experiments have binary sensitive attributes, as the NIFA paper is also focussed on binary sensitive attributes.
>
>
> >7. Luo et al. show that increasing the removal of uncertain nodes (via η) can improve fairness with minimal utility loss. >Accordingly, it seems that removing uncertain nodes can reduce the fairness metrics, but I wonder if the author can discuss it >further in the context of previous work, although the attack type is different.
>
> Response: Thank you for this addition, we have improved our comparison by adding results for masking defense, and node defense on FA-GNN, FATE, and GFair attacks (Section 4.3, table 8). We observe good generalizability of the Masking defense.
>
>
> >8. The statement “evaluating the poisoned model on a clean train graph significantly reduces the impact of NIFA” is potentially misleading. It seems that “train” was intended to mean “test” in this context. Please clarify the intended meaning
>
> Response: Thank you for pointing this out. You are correct, we intended to say 'test' instead of 'train.' We have corrected this in the revision.

---

> > ### Author Response · Authors · 2025-04-01
> > **Author response to Weaknesses**
> >
> > > Weaknesses:
> > 1. The authors redefine NIFA as an evasion attack, yet do not provide sufficient theoretical justification for how this diverges from its original categorization as a poisoning attack. Since the two attack types differ fundamentally in >terms of when and how the injected nodes influence the model (training-time vs. inference-time effects), a more >formal and explicit clarification would help avoid conceptual confusion.
> >
> > Response: In section 4.3 we have provided an explanation. If you think a theoretical and formal explanation of this concept is necessary, we are willing to revise this section.
> >
> >
> > >2. In Section 5.1 and Appendix C, the authors report that a “true poisoning attack” (i.e., poisoning only the training graph) leads to a reduction in fairness degradation relative to the full NIFA attack. This directly challenges the claim that NIFA consistently undermines fairness and may leave readers unclear about the actual mechanism through wich >NIFA affects model fairness.
> >
> > Response: Indeed, it is true that when modelled as a true poisoning attack, NIFA actually does not undermine fairness. As an explanation, we have added the following to Section 5.1:
> >
> > These results strengthen our hypothesis about NIFA impacting fairness through the features propagated from the poisoned nodes, instead of the poisoned model itself. It could indicate limited effectiveness of feature-based poisoning attack analogous to NIFA. This is crucial, as it tells us that on one hand, poisoning a model with such an attack has limited effectiveness, while at the same time, existing malicious nodes can have a big impact during inference. We further hypothesize that injecting poison nodes into the training set introduces invariance in the victim model towards the attack. Future work could thus study NIFA as a fair training technique.
> >
> >
> > >3. While the proposed Antidote Defense mirrors the NIFA optimization pipeline in the opposite direction, it remains unclear why such a defense should generalize beyond NIFA to other fairness attacks (e.g., FA-GNN, FATE, G-FairAttack). Either theoretical reasoning or additional experiments would strengthen the case for the defense's general applicability.
> >
> > To illustrate the effects of the antidote defense when used with other fairness attacks, we performed additional experiments of Antidote defense and Masking defense on the other attack methods. The results can be found in Section 4.3, table 8.
> >
> >
> > >4. In several cases — particularly on the Pokec dataset — the application of NIFA appears to improve fairness metrics (ΔSP and ΔEO). If the attack does not consistently degrade fairness, the stealthiness and effectiveness of NIFA as a fairness attack become less convincing. A more detailed discussion of such failure or edge cases would help clarify the scope and limitations of the attack.
> >
> > Response: In case of FairVGNN, on the Pokec dataset, NIFA does fail to work for certain hyperparameter values. To explain this, we have added the following to section 5.2 (Anomalous FairVGNN results on Pokec):
> > These results highlight the sensitivity of NIFA to hyperparameters, suggesting that its effectiveness is more variable than the paper indicates. Apart from that, it also shows the potential of limiting sensitive attribute leakage as a defense mechanism against a fairness attack like NIFA. Future work could look at other methods similar to FairVGNN that work on sensitive attribute leakage.
> > In addition, we have added an explanation as well for how NIFA works in 5.1 (NIFA as a true poisoning attack), mentioned in one of the earlier comments.
> >
> >
> > >5. On the DBLP dataset, the observed accuracy drops from 3% to over 5%, depending on the model and attack mode (e.g., evasion). These changes are significant and suggest that the claim of NIFA having no utility impact may be overstated. It would be more accurate to qualify the extent of the utility-preserving property.
> > We appreciate the sharp perspective. In response we have added this point in Section 4.1.

---

> ### Comment · Reviewer_8vZT · 2025-04-04
>
> Thank you for addressing many of my questions. However, I still have some questions and minor comments:
>
> Q1. Table 8 shows that the Antidote defense actually worsens fairness metrics in some cases (e.g., FATE + Antidote on DBLP). Could you analyze the causes of these cases and elaborate on the limitations of the Antidote defense?
>
> Q2. In Figure 10, an opposite phenomenon is observed in the DBLP dataset where the standard deviation increases when adding layers. How does this difference affect the fairness metrics?
>
> Q3. If over-smoothing acts as a natural defense mechanism against fairness attacks, would it be possible to intentionally leverage this mechanism and compare it experimentally with the proposed Antidote defense?
>
> Q4. In the content added to Section 5.1, you claim that NIFA's fairness-harming effects come primarily from contaminated nodes during the inference phase, while simultaneously mentioning that "poisoned nodes into the training set introduces invariance in the victim model towards the attack." How do these two mechanisms operate simultaneously? Could you elaborate on how the model gains "invariance" during the training phase while contaminated nodes still have a significant impact on fairness during the inference phase? Is my understanding correct?
>
> Minor comments:
> - There is a typo "indidual" in the first paragraph of Appendix H.1.
> - It would be helpful for reviewers if the revised manuscript highlighted which additions and modifications correspond to which reviewer's feedback.

---

> > ### Author Response · Authors · 2025-04-10
> > **Author Response to Additional Questions**
> >
> > >Q1. Table 8 shows that the Antidote defense actually worsens fairness metrics in some cases (e.g., FATE + Antidote on DBLP). Could you analyze the causes of these cases and elaborate on the limitations of the Antidote defense?
> >
> > Response: We believe that the reason the defense is ineffective on FATE and FA-GNN is related partly to the attack methods and can partly be attributed to the properties of the graph. We say the former because we see that on G-Fair, while not ideal, it does manage to mitigate the effectiveness of the attack. However, all three attacks work based on edges, and sparser graphs might limit the effectiveness of feature propagation from our good nodes. We mention sparsity because FA-GNN works well on the Pokec datasets.
> >
> > >Q2. In Figure 10, an opposite phenomenon is observed in the DBLP dataset where the standard deviation increases when adding layers. How does this difference affect the fairness metrics?
> >
> > Response: We think that DBLP’s sparsity has a role to play in this phenomenon. Average node degree is smaller in DBLP compared to the pokec datasets, and neighborhood increases exponentially with every layer. Since DBLP is sparse, this remains much smaller even after multiple layers. However, we do not have a certain answer as of now.
> >
> > >Q3. If over-smoothing acts as a natural defense mechanism against fairness attacks, would it be possible to intentionally leverage this mechanism and compare it experimentally with the proposed Antidote defense?
> >
> > Response: We do not think it is oversmoothing that acts as the defense but rather increase of feature propagation depth. Oversmoothing is just a phenomenon that gets more prominent with deeper layers, and the degradation we see after some improvement in fairness is likely because the effects of oversmoothing outweigh the gains of propagation depth.
> >
> > >Q4. In the content added to Section 5.1, you claim that NIFA's fairness-harming effects come primarily from contaminated nodes during the inference phase, while simultaneously mentioning that "poisoned nodes into the training set introduces invariance in the victim model towards the attack." How do these two mechanisms operate simultaneously? Could you elaborate on how the model gains "invariance" during the training phase while contaminated nodes still have a significant impact on fairness during the inference phase? Is my understanding correct?
> >
> > Response: Since our evasion attack works much better than the original setting, and NIFA does not work as a true poisoning attack, we reason that NIFA actually works by feature propagation. Through feature propagation, NIFA makes the fairness worse during evaluation, but having NIFA during training allows the model to counter it to an extent. This is the setting of the true poisoning attack (Appendix D), where NIFA is used during training but evaluation is done on a clean graph. We see in absence of the malicious nodes, not only does the fairness not worsen, but also that it actually improves because of training on the attacked graph.
> >
> > Minor comments:
> > >There is a typo "indidual" in the first paragraph of Appendix H.1.
> >
> > Thank you for pointing this out. We have corrected the mistake.
> >
> > >It would be helpful for reviewers if the revised manuscript highlighted which additions and modifications correspond to which reviewer's feedback.
> >
> > Thanks for the suggestion. We have underlined the changes and added notes alongside.

---

> ### Comment · Reviewer_8vZT · 2025-04-11
>
> I thank the authors for their responses. I believe most of the previous questions are addressed. However, I would like to request further clarification on one point:
>
> In your response to Q2, the authors mention DBLP's sparsity characteristics. I also found the sentence in Appendix H: "This might be due to the sparseness of the graph compared to the Pokec datasets."
> While the authors' statement can be derived from the average node degrees in Table 3, this provides only a high-level view of the graph structure. Could you analyze the relationship between nodes with low degree versus high degree groups to better support this hypothesis? Perhaps checking how fairness metrics vary on different degree ranges would provide better evidence for your explanation.
>
> Minor comment:
>
> There appears to be a notation overlap for '$d$' between Table 2 and Table 3, which might confuse readers.

---

> > ### Author Response · Authors · 2025-04-11
> > **Author Response to Comment**
> >
> > Thank you for your suggestion.
> >
> > > Could you analyze the relationship between nodes with low degree versus high degree groups to better support this hypothesis? Perhaps checking how fairness metrics vary on different degree ranges would provide better evidence for your explanation.
> >
> > Response:
> > We have added additional results to the appendix section D to substantiate our claims. We found that a graph with a lower degree will most likely exacerbate the effectiveness of NIFA. A high degree graph would result in a less significant impact of injected nodes on aggregated features during node classification.

---

> ### Comment · Reviewer_8vZT · 2025-04-15
>
> Thank you for your response. My previous concerns have been addressed.

---

### Review · Reviewer_AtBc · 2025-03-27

**Summary Of Contributions:**

This paper presents a reproducibility study of Luo et al. (2024), which introduces a node injection-based fairness attack (NIFA) on graph neural networks. The authors successfully reproduce the original results and extend the work with three additional experiments: (1) evaluating NIFA as an evasion attack, (2) proposing a novel defense strategy called Antidote Defense, and (3) investigating the impact of model depth on attack effectiveness.

**Audience:**

Yes

**Claims And Evidence:**

No

**Requested Changes:**

Given the current state of the manuscript, I recommend rejection, with encouragement for the authors to substantially expand their work before resubmission. Specifically:

- Develop more sophisticated defense mechanisms with stronger theoretical foundations, rather than simply inverting the attack method.
- Provide deeper theoretical analysis explaining the observed phenomena, particularly regarding why NIFA works as an evasion attack and why model depth affects fairness vulnerability.
-  Expand the experimental evaluation to additional datasets and more recent GNN architectures to demonstrate broader applicability.
-  Investigate the anomalous results with FairVGNN more thoroughly, as these could reveal important insights about fairness-aware models.
-  Create original visualizations and diagrams rather than relying on those from the original paper, to better demonstrate independent understanding of the mechanisms.

**Strengths And Weaknesses:**

**Strengths**:

- The reproducibility aspect is thorough, with clear verification of the original paper's claims through carefully designed experiments.
- The paper is well-structured and clearly written, with detailed explanations of methodologies and results.
- The proposed extensions, particularly the exploration of NIFA as an evasion attack, offer interesting insights into the attack mechanism.
- The authors' implementation of a defense strategy represents a valuable contribution beyond mere reproduction.


**Weaknesses**:
- Limited originality and insufficient novelty: While reproducibility studies are valuable, the paper's extensions lack substantial innovation. The evasion attack is primarily a reframing of the original attack rather than a novel contribution. TMLR typically expects journal submissions to provide significant new content (approximately 30% new material) compared to conference papers.
- Simple defense strategy: The "Antidote Defense" is essentially an inverse application of NIFA's own principles. This represents a straightforward approach rather than a sophisticated defense mechanism with theoretical guarantees or novel insights.
- Superficial analysis of model depth experiments: The exploration of deeper models, while potentially interesting, lacks theoretical grounding and in-depth analysis of why certain depth configurations mitigate fairness degradation.
- Limited experimental diversity: The extensions use the same datasets and largely the same models as the original paper, missing an opportunity to test generalizability across diverse graph structures or newer GNN architectures.
- Anomalous results unexplained: The inconsistent findings with FairVGNN on Pokec datasets are noted but not sufficiently investigated or explained from a theoretical perspective.
- Heavy reliance on original paper's figures and framework: The paper directly reuses Figure 1 from the original paper without sufficient modification or original visualization contributions. This further emphasizes the limited novelty of the work and suggests over-reliance on the original authors' explanations rather than developing a unique perspective.

---

> ### Author Response · Authors · 2025-04-10
> **Author Response to Requested Changes**
>
> Thank you for your feedback — we appreciate your insights and your help in improving our study.
>
> > Develop more sophisticated defense mechanisms with stronger theoretical foundations, rather than simply inverting the attack method.
>
> Response: We appreciate the suggestion, and agree that the defense mechanism is quite simple. However, we believe that the similarity to the attack method gives our defense a theoretical grounding in the principles of the attack. It also makes it intuitive, since it directly counters the unfairness induced by the attack. Additionally, our experiments show that the defense counters more than just NIFA, despite the simplicity. Therefore, we believe that the strength of the approach lies in its simplicity, which enables strong empirical performance without the need for added complexity.
>
> >Provide deeper theoretical analysis explaining the observed phenomena, particularly regarding why NIFA works as an evasion attack and why model depth affects fairness vulnerability.
>
> Response: Thank you for this suggestion. To increase the robustness of our research, we have added a potential explanation of the Layer experiments in Section 5.3, and Appendix H.2. We found that increasing the amount of layers in the victim model increases the amount of oversmoothing in output features of the penultimate layer, before the activation. This increased oversmoothing correlates with the decrease in Delta EO and Delta SP, showing that there might be a causal relationship between the two. We have added several improvements to the overall clarity of our research, such as improving our figures and expanding our arguments in section 5.
>
> >Expand the experimental evaluation to additional datasets and more recent GNN architectures to demonstrate broader applicability.
>
> Response: We have added an additional model, GAT, to the main results in Table 5, and evasion results in Table 6. The GAT model encompasses an architecture which seems quite invariant to vanilla NIFA, but suffers significantly from the evasion version. One reason we did not add many more architectures is that the original paper is new and uses fairly recent architectures.
> Regarding the datasets, we chose not to expand beyond the 3 datasets (Pokec-n, Pokec-z, and DBLP) because they are representative and commonly used in related studies. While we agree that including additional datasets could further support generalizability, this would significantly increase computational costs and strain the available space for detailed analysis. We believe our current selection strikes a good balance between demonstrating generalizability and enabling in-depth evaluation.
>
>
> >Investigate the anomalous results with FairVGNN more thoroughly, as these could reveal important insights about fairness-aware models.
>
> Response: We do agree that some explanation was lacking. We have added additional reasoning to Section 5.2.
>
> >Create original visualizations and diagrams rather than relying on those from the original paper, to better demonstrate independent understanding of the mechanisms
>
> Response: We have added an augmented version of Figure 1. We agree that the tables for results in our research follow the exact formatting as that of the original research. This was done for easy comparison, and to keep a consistent and interpretable paper. Utilizing different visualizations for our research would not add to the research, but does reduce this ease of comparison. We hope that this augmented version strikes a good balance between familiarity to the NIFA paper and detail of our added mechanisms.

---

### Decision · Action_Editor_DpMZ · 2025-05-12

**Recommendation:** Reject

**Comment:**

Although this paper presents a comprehensive reproducibility study, reviewers raised some critical concerns. For example, the original paper of NIFA already provided source codes, and thus the contribution of this reproducibility study is not quite clear. Also, the new insights presented in this paper are not sufficient. Overall, this paper is not ready for publication at TMLR.

**Audience:**

TMLR's audience might not be interested in this reproducibility study, as the original paper already provides implementation details and source codes.

**Claims And Evidence:**

This paper presents a reproducibility study of node injection-based fairness attack (NIFA) on graph neural networks, proposed by Luo et al. in 2024. Although the claims and evidences are mostly consistent with the original paper of NIFA, this paper does not provide insufficient new insights or new findings.
In the rebuttal, the authors claimed that "However, since the code for fairness-aware models was not included in the repository of the paper, we had to refer to the original papers for the attacks". In fact, in the GitHub repository of NIFA (https://github.com/CGCL-codes/NIFA), the authors of NIFA already explained why the fairness-aware models were not included in the NIFA repository. And the authors of NIFA also provided details on evaluating the fairness-aware models.